# Circulating tumor DNA refines risk stratification of neoadjuvant therapy-resistant breast tumors

Mark Jesus M. Magbanua [1] ✉, Nayelis A. Manon[1], Denise M. Wolf[1], Samuel Rivero-Hinojosa[2], Ziad Ahmed[1], Rosalyn W. Sayaman[1], Antony Tin[2], Derrick Renner[2], Ekaterina Kalashnikova[2], Lamorna Brown-Swigart[1], Gillian L. Hirst[3], Christina Yau[3], Wen Li[4], Claudine Isaacs[5], Rebecca A. Shatsky[6], Amy S. Clark[7], Alexandra Zimmer[8], Amy L. Delson[9], Angel Rodriguez[2], Minetta C. Liu[2], Paula R. Pohlmann[10], Laura J. Esserman[3], Hope S. Rugo[11], Angela DeMichele[7] & Laura van 't Veer[1]

Early-stage breast cancers resistant to neoadjuvant therapy (NAT), characterized by high residual cancer burden (RCB) after treatment, have an increased risk of metastatic recurrence. Here, we show that circulating tumor DNA (ctDNA) detected using a tumor-informed test (1) can improve risk stratification of patients with NAT-resistant tumors (RCB-II/RCB-III) and (2) predict response to NAT. Stratification using ctDNA status at pretreatment or post-NAT and ctDNA dynamics identified NAT-resistant tumors with a significantly decreased risk of metastatic recurrence. ctDNA clearance as early as week 3 across receptor subtypes predicted favorable responses to NAT, including immunotherapies. Interestingly, less than a fifth of patients with NAT-resistant tumors were ctDNA-positive post-NAT. Serial mutation profiling of NAT-resistant tumors revealed that patient-specific ctDNA assay variants remained detectable over time, including in tumors of patients ctDNA-negative post-NAT. Refining risk stratification for NAT-resistant tumors using ctDNA and understanding ctDNA shedding in these tumors could guide treatment decisions to prevent or delay metastatic recurrence.

Metastatic breast cancer is a major driver of morbidity and mortality, and thus, preventing metastatic recurrence is crucial for improving patient outcomes. An approach to reducing the risk of the cancer spreading is to treat breast tumors before surgery with neoadjuvant therapy (NAT)[1,2]. A pathologic complete response (pCR) to NAT, marked by the absence of residual cancer burden (RCB-0) in the breast and regional lymph nodes, is a strong predictor of favorable long-term outcomes, especially in HER2-positive and triple-negative (TN) breast

[1]Department of Laboratory Medicine, University of California San Francisco, San Francisco, CA, USA. [2]Natera Inc., Austin, TX, USA. [3]Department of Surgery, University of California San Francisco, San Francisco, CA, USA. [4]Department of Radiology, University of California San Francisco, San Francisco, CA, USA. [5]Lombardi Comprehensive Cancer Center, Georgetown University Medical Center, Washington, DC, USA. [6]Department of Medicine, University of California San Diego, La Jolla, CA, USA. [7]Division of Hematology/Oncology, University of Pennsylvania, Philadelphia, PA, USA. [8]Division of Hematology/Oncology, Oregon Health and Science University, Portland, OR, USA. [9]Breast Science Advocacy Core, University of California San Francisco, San Francisco, CA, USA. [10]Department of Breast Medical Oncology, University of Texas MD Anderson Cancer Center, Houston, TX, USA. [11]Division of Hematology/Oncology, University of California San Francisco, San Francisco, CA, USA. ✉e-mail: mark.magbanua@ucsf.edu

cancer subtypes; however, its prognostic value is less robust in hormone receptor (HR)-positive/HER2-negative disease[3]. Achieving a pCR/RCB-0 or having a limited residual cancer burden (RCB-I) has a 3-year distant recurrence-free survival (DRFS) rate of greater than 90%[4–6]. However, about 50% of patients have tumors that are resistant to NAT[6,7], defined as having moderate (RCB-II) or extensive (RCB-III) invasive cancer after treatment. An RCB-II/RCB-III diagnosis poorly predicts metastatic recurrence despite a large tumor burden after NAT. Only about 15–30% of patients with NAT-resistant tumors experience a metastatic recurrence within 3 years of follow-up[4–6]. This highlights the heterogeneity of NAT-resistant tumors and their propensity to metastasize, underscoring the importance of accurate risk stratification for patients with RCB-II/RCB-III.

Monitoring tumor response during NAT to guide therapeutic decisions on whether to continue or switch therapies (treatment redirection) can increase the likelihood of a favorable response[8,9]. However, repeat tissue biopsies pose significant risks and are uncomfortable for the patient. Utilizing less invasive methods, such as liquid biopsy, can enable serial evaluations with minimal risk. Circulating tumor DNA (ctDNA) is a promising liquid biopsy biomarker for identifying non-responding tumors during NAT and those with an increased risk of relapse[10–16]. Clinical studies from our group[10–12] and others[17,18] have demonstrated the potential utility of ctDNA for monitoring treatment response and predicting metastatic recurrence in patients with early-stage breast cancer. Using ctDNA information for early identification of non-responders and those at risk of metastatic recurrence can provide opportunities for more aggressive treatment (escalation) to delay or prevent metastatic spread. Conversely, among patients whose NAT-resistant tumors have lower metastatic risk, less aggressive treatment (de-escalation) can reduce exposure to the toxicities of unnecessary therapies.

There are various types of ctDNA assays designed for different applications[17,19]. The main categories include tumor-agnostic and tumor-informed tests[20]. Tumor-agnostic methods do not require tumor sequencing and utilize the same fixed panel of assays in every case to detect common cancer mutations in the blood[21]. In contrast, tumor-informed methods need prior knowledge of existing tumor mutations; thus, sequencing of the tumor tissue is a prerequisite to "inform" the subsequent design of patient-specific ctDNA assays[21]. The sequence information, however, only provides a snapshot of the tumor's molecular profile at a particular point in time. Changes in the mutational landscape during tumor evolution −whether due to clonal selection, therapeutic pressure, or spatial heterogeneity−can result in the emergence of mutations that are not present in the profiled tumor sample.

Breast cancer is a heterogeneous disease defined by receptor subtypes based on hormone receptor (HR) and HER2 status, each with different sensitivities to NAT[22] and survival outcomes[23,24]. Considering this heterogeneity, we recently compared the clinical and biological correlates of ctDNA in HER2-negative subtypes: triple-negative (TN, $n = 138$) versus HR-positive/HER2-negative ($n = 145$)[11].

Here, we report the findings from our expanded ctDNA study of 723 patients, including HER2-positive breast cancers. We used a tumor-informed, personalized ctDNA test to detect patient-specific tumor variants in the blood[11,12]. This study examined whether ctDNA could improve prognostication using RCB class[4–6] and predict response to NAT, including treatment containing drugs that target checkpoint proteins (immunotherapies). Finally, we analyzed serial tumor mutational profiling data from matched NAT-resistant tumors to investigate whether changes in the mutational landscape during NAT impacted the detection of patient-specific ctDNA assay variants.

## Results

### Patients, ctDNA testing, and correlative tissue studies
We performed ctDNA analysis in patients with high-risk (MammaPrint high) early-stage breast cancer receiving NAT in the I-SPY2 trial

(Fig. 1A). Of the 723 evaluable patients, 300 (41%) were HR-positive/HER2-negative, 237 (33%) were TN, and 186 (26%) were HER2-positive (Fig. 1B). The clinicopathologic characteristics of the patient cohort are summarized in Table S1.

We used a tumor-informed, personalized assay to detect ctDNA in blood collected at 4 time points: at pretreatment (T0), 3 weeks after initiation of paclitaxel treatment with or without investigational agents (T1), 12 weeks between paclitaxel-based and anthracycline treatment (T2), and after NAT (T3). ctDNA data were generated in 2607 samples (Fig. 1A). The analytic pipelines, including correlative studies, are outlined in Fig. 1C.

### ctDNA status and trajectory across receptor subtypes
The ctDNA positivity rate at pretreatment (T0) was highest in the TN subtype (92%) (Fig. 2A), consistent with previous findings[11,12]. There was no significant difference in the pretreatment (T0) ctDNA positivity rates observed between the HR-positive/HER2-negative and the HER2-positive groups (76% versus 77%, Chi-squared ($\chi^2$) $p = 0.75$). However, 3 weeks after initiation of treatment (T1), the ctDNA positivity rate in the HR-positive/HER2-negative subtype was significantly higher compared to that of the HER2-positive group (45% versus 25%, $\chi^2$ $p < 0.001$). The ctDNA positivity rate remained the lowest in the HER2-positive group at 12 weeks (T2) and post-NAT before surgery (T3) (Fig. 2B).

### ctDNA independently predicts metastatic recurrence
Next, we assessed the prognostic significance of ctDNA, along with other clinicopathological factors, including clinical T and N stages, grade, MammaPrint score, and RCB. DRFS data were available for 712 patients, of whom 133 (19%) experienced a DRFS event. The median follow-up was 4.7 years.

Multivariable Cox regression analysis, which included clinicopathologic variables that were statistically significant in univariable analyses (Table S2), showed that ctDNA positivity at pretreatment (T0) is an independent predictor of metastatic recurrence [adjusted hazard ratio (adj HzR) = 4.40, 95% confidence interval (CI) 1.91–10.16, Wald $p = 0.001$] (Fig. S1A). The same analysis also identified ctDNA positivity after NAT before surgery (T3) as an independent prognostic factor for poor outcomes [adj HzR=5.20, 95% CI 3.24–8.35, Wald $p < 0.001$] (Fig. S1B).

We then assessed the prognostic significance of ctDNA dynamics (timing of ctDNA clearance). Patients were divided into 5 groups: those with persistently ctDNA-negative tests, those with ctDNA clearance at week 3 (T1), week 12 (T2), or post-NAT before surgery (T3), and those with no ctDNA clearance (no clearance at T3). Late and no clearance were both significant independent negative prognostic factors for DRFS [adj HzR=6.88, 95% CI 2.37–19.98, Wald $p < 0.001$; adj HzR=16.50, 95% CI 5.67–47.98, Wald $p < 0.001$, respectively] (Fig. S1C).

### ctDNA improves risk stratification of NAT-resistant tumors
A total of 708 patients had RCB data, of whom 321 (45%) had NAT-responsive tumors (RCB-0/RCB-I). Of these, 702 had DRFS data. Expectedly, we observed a high specificity of RCB-0 or RCB-I for predicting favorable survival. The 3-year DRFS rates in patients with RCB-0 and RCB-I were 94% and 89%, respectively (Fig. 3A).

Of the 708 patients with RCB data, 55% had NAT-resistant tumors: 255 RCB-II (36%) and 132 RCB-III (19%). In the 702 patients with DRFS data, an RCB-II/RCB-III after NAT was significantly associated with worse DRFS (log-rank $p < 0.0001$) (Fig. 3A). However, despite the high tumor burden after NAT, only 17% (RCB-II) and 33% (RCB-III) of patients experienced metastatic recurrence or death after 3 years of follow-up. Due to the low specificity of RCB-II/RCB-III in predicting DRFS, we investigated whether ctDNA can refine risk stratification in patients with NAT-resistant tumors.

First, we stratified patients with NAT-resistant tumors (RCB-II and RCB-III) by ctDNA status (ctDNA-positive versus ctDNA-negative) at

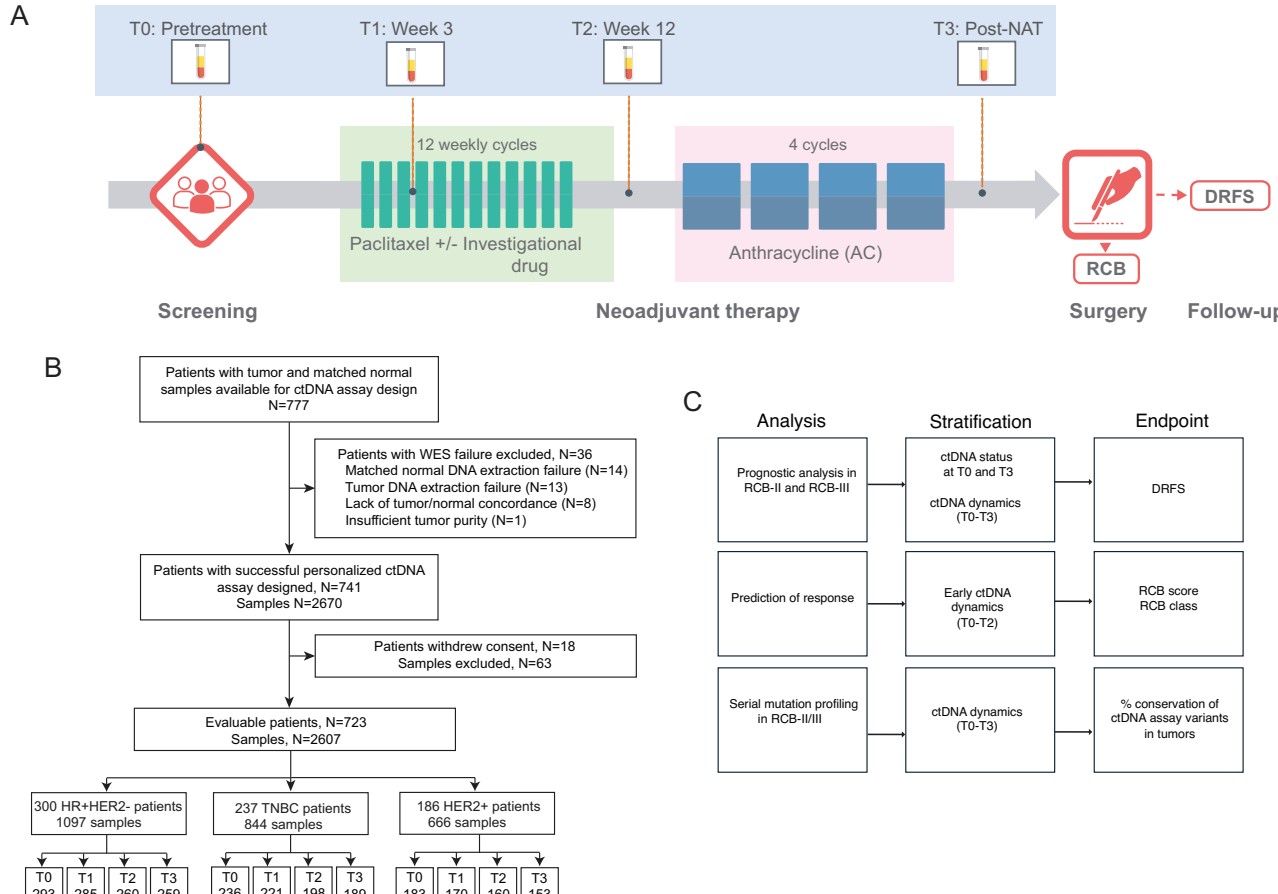

**Fig. 1 | ctDNA analysis in high-risk early-stage breast cancer receiving neoadjuvant therapy (NAT). A** Blood samples for ctDNA analysis were collected at pretreatment (T0), 3 weeks after initiation of paclitaxel treatment with or without an investigational drug (T1), at 12 weeks post-paclitaxel treatment before the anthracycline (AC) regimen (T2), and post-NAT before surgery (T3). The survival endpoint was distant recurrence-free survival (DRFS), and the response endpoints were RCB score and RCB class. **B** Inclusion or exclusion of patients and samples from the analyses based on evaluability and quality control metrics. **C** Analytic pipeline for clinical and correlative studies.

pretreatment (T0) and compared DRFS between groups. Patients who tested ctDNA-negative at pretreatment (T0) had a significantly higher 3-year DRFS rate versus those who tested ctDNA-positive in both RCB-II [98% versus 81%, adj HzR=0.10, 95% CI 0.01–0.73, Wald $p = 0.02$] and RCB-III [94% versus 59%, adj HzR=0.28, 95% CI 0.09–0.92, Wald $p = 0.04$] groups (Fig. 3B, Fig. S2A). The same analysis using ctDNA status post-NAT before surgery (T3) yielded similar but more statistically significant differences in DRFS (Fig. 3C, Fig. S2B): ctDNA-negativity was associated with significantly improved 3-year DRFS rates in patients with RCB-II (88% versus 57%; adj HzR=0.29, 95% CI 0.13-0.61, Wald $p = 0.001$) and RCB-III (83% versus 22%; adj HzR=0.14, 95% CI 0.07-0.26, Wald $p < 0.001$). These results demonstrate that ctDNA negativity at pretreatment (T0) and post-NAT before surgery (T3) significantly correlated with reduced risk of metastatic recurrence, even in patients with RCB-II or RCB-III after NAT.

Next, we grouped patients by ctDNA dynamics. Since this analysis included only patients with complete ctDNA for all 4 time points, we combined the analyses for patients with RCB-II and RCB-III to increase the sample size ($n = 249$). Patients with persistently ctDNA-negative tests (98%), or with early ctDNA clearance at week 3 (T1, 91%) or week 12 (T2, 92%) had significantly higher 3-year DRFS rates compared to those with late ctDNA clearance at post-NAT (T3, 63%) or those with no ctDNA clearance (35%, log-rank $p < 0.0001$, Fig. 3D, Fig. S3A). Similar results were observed across receptor subtypes (Fig. S3B).

We assessed the prognostic performance of the survival (Cox) models for predicting DRFS. To compare model performance, we analyzed survival data from patients common to all the Cox models ($n = 249$). Evaluation of the prognostic performance of the RCB class model (RCB-II vs. RCB-III in Fig. 3A) revealed a Harrell's c-index of 0.75 (see Methods, Fig. 3E, Table 1). Numerically higher c-indices were observed in models that include ctDNA information with RCB-II/III as predictors of DRFS (Fig. 3B–D).

Taken together, our results indicate that ctDNA can fine-tune risk stratification in patients with RCB-II/RCB-III by identifying those with increased metastatic risk, i.e., ctDNA-positive at pretreatment (T0) or post-NAT (T3) or no ctDNA clearance at T3, versus those with a reduced propensity to metastasize, i.e., persistently ctDNA-negative and with early ctDNA clearance. Similar survival analyses performed in patients with RCB-0 or RCB-I also identified patients with poor survival, e.g., those who were ctDNA-positive post-NAT (T3) or those with no ctDNA clearance (Fig. S4).

**Predictive performance of the ctDNA assay**
We assessed the predictive performance of the ctDNA assay by measuring the test's positive predictive value (PPV) and sensitivity in correctly identifying patients with RCB-II/III, as well as the negative predictive value (NPV) and specificity in correctly identifying patients with RCB-0/I.

The analysis revealed increasing PPV across time points for all receptor subtypes, but not for NPV (Fig. 4A, Fig. S5). Higher PPV and lower NPV at all time points were observed in the HR-positive/HER2-negative subtype compared to the other receptor subtypes.

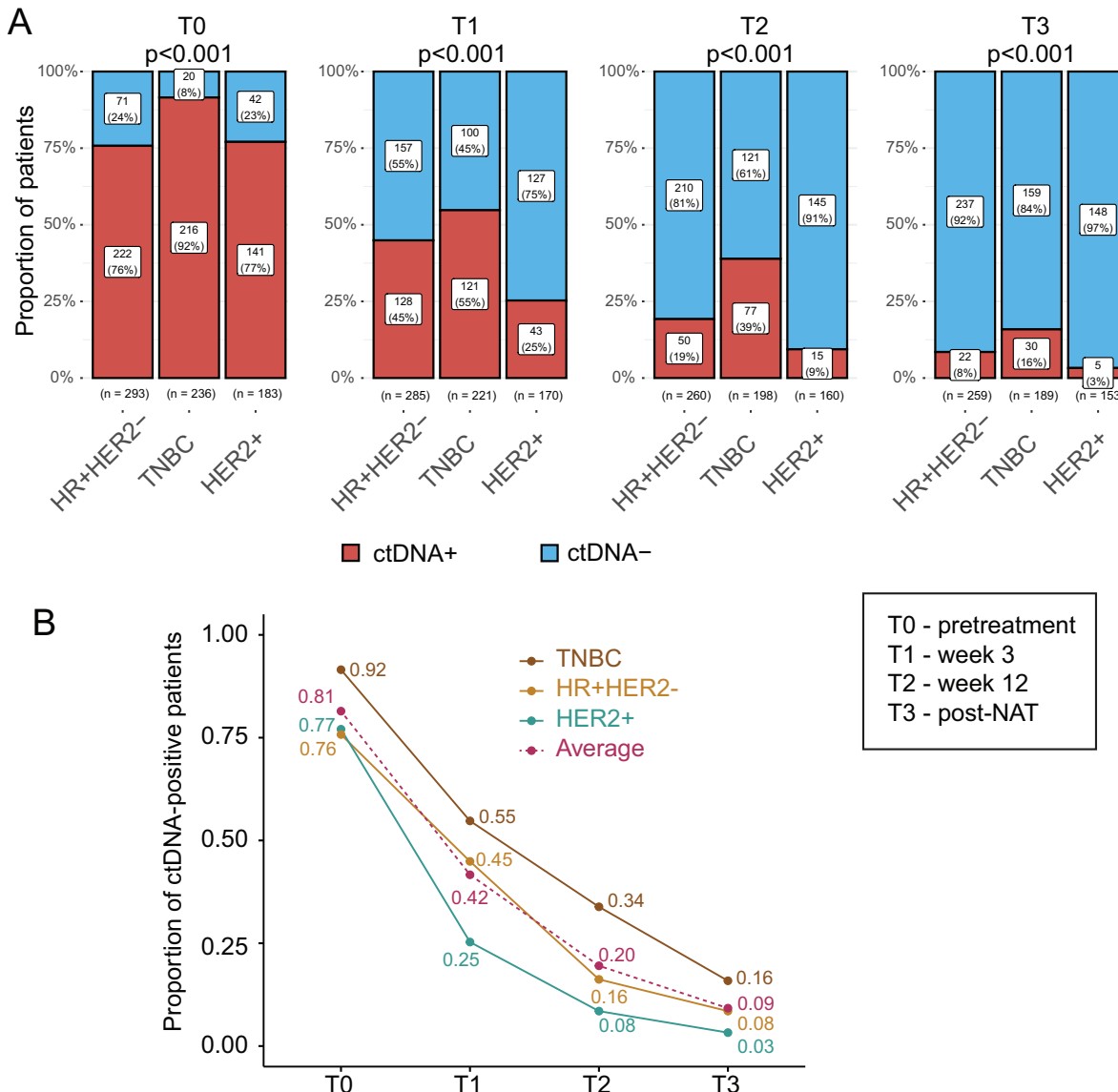

**Fig. 2 | ctDNA positivity before, during, and after neoadjuvant therapy across receptor subtypes. A** Bar plots showing the proportion of patients by the ctDNA status (ctDNA-positive versus ctDNA-negative) at pretreatment (T0), 3 weeks after initiation of paclitaxel treatment with or without an investigational drug (T1), at 12 weeks post-paclitaxel treatment before the anthracycline (AC) regimen (T2), and post-NAT before surgery (T3) in hormone receptor-positive/HER2-negative (HR + HER2-), triple-negative breast cancer (TNBC), and HER2-positive (HER2 + ) breast cancer subtypes. The *p*-values were calculated using Chi-squared tests. **B** Line plot showing the proportion of ctDNA-positive patients from T0 to T3, stratified by receptor subtype.

Furthermore, the sensitivity of the test diminished over time across all receptor subtypes, whereas its specificity improved. Sensitivity was highest in the TN group, while specificity was comparable across all receptor subtypes.

### Early ctDNA clearance is associated with a favorable response to NAT

Next, we evaluated the predictive value of ctDNA using the RCB score (the continuous measure of RCB) and the RCB class as the response endpoints. Given the known differences in response rates to NAT across receptor subtypes, we performed the analysis within each group.

Patients were assigned to one of the 4 groups based on early ctDNA dynamics: persistently ctDNA-negative, cleared ctDNA at week 3 (T1) or 12 (T2), or no clearance at T2. Post-NAT (T3) ctDNA data were excluded from the stratification due to their temporal proximity to surgery when RCB is evaluated. Consistent with the highest response

rates across receptor subtypes (Table S1), the HER2-positive group had the largest proportion of patients with ctDNA clearance at week 3 (T1), at 57%, compared to 32% in the HR-positive/HER2-negative and TN groups (Fig. 4B).

We examined whether early ctDNA dynamics from pretreatment (T0) to week 3 (T1) and 12 (T2) were associated with resistance (RCB-II/ RCB-III) or sensitivity (RCB-0/RCB-I) to NAT. Our analysis revealed that ctDNA clearance at week 3 (T1) was significantly associated with a favorable response. The proportion of patients with an RCB-0/RCB-I after NAT was highest among those with early ctDNA clearance at week 3 (T1) (Fig. 4C). This pattern was consistently observed across all receptor subtypes [HR-positive/HER2-negative: 42%, $\chi^2$ *p* = 0.02; TN: 82%, $\chi^2$ *p* < 0.001; HER2-positive: 82% $\chi^2$ *p* < 0.001]. Significant associations were observed between early clearance at week 3 (T1) and a favorable response to NAT, regardless of clinical T stage or nodal status (all $\chi^2$, *p* < 0.05; Fig. S6). In addition, early ctDNA clearance at week 3 (T1) was associated with a significantly increased likelihood of

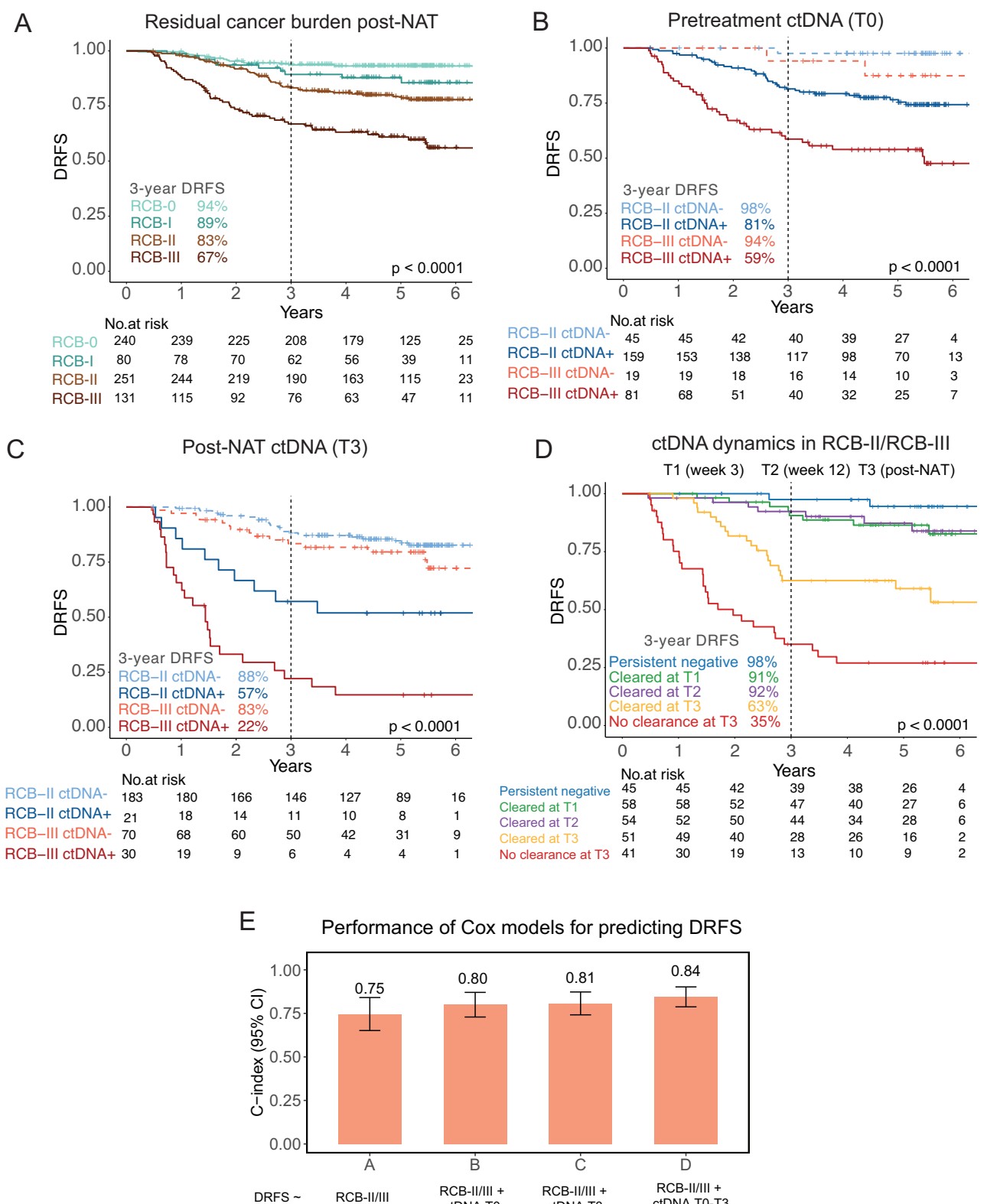

**Fig. 3 | ctDNA predicts metastatic recurrence and refines risk stratification of breast tumors resistant to neoadjuvant therapy (NAT).** Kaplan–Meier estimates for 3-year distant recurrence-free survival (DRFS) rates were calculated in **A** patients stratified by residual cancer burden (RCB) class: RCB-0 (pathologic complete response), RCB-I (limited), RCB-II (moderate), and RCB-III (extensive); in patients with NAT-resistant tumors defined as RCB-II/RCB-III stratified by **B** ctDNA status at pretreatment (T0) or **C** ctDNA status post-NAT before surgery (T3); and **D** in all patients with RCB-II/RCB-III stratified by ctDNA dynamics or the timing of ctDNA clearance. Patients were grouped into persistent ctDNA-negative, ctDNA cleared at week 3 (T1), week 12 (T2), or post-NAT before surgery (T3), and no ctDNA clearance post-NAT before surgery. The *p*-values for the survival curves were calculated using the log-rank test. **E** Model discrimination was assessed using Harrell's concordance index (c-index) to evaluate the prognostic performance of the risk models for predicting DRFS. The error bars represent the 95% confidence intervals (see Table 1).

**Table 1 | Model discrimination using Harrell's concordance index (c-index) to evaluate the prognostic performance of the risk models for predicting distant recurrence-free survival (DRFS)**

| Cox models | A | B | C | D |
|---|---|---|---|---|
| DRFS ~ | RCB-II/III | RCB-II/III + ctDNA T0 | RCB-II/III + ctDNA T3 | RCB-II/III + ctDNA dynamics (T0-T3) |
| Predictors | RCB-II vs. RCB-III | RCB – II/ctDNA+ vs. RCB-II/ctDNA- vs. RCB-III/ctDNA+ vs. RCB-III/ctDNA- | RCB – II/ctDNA+ vs. RCB-II/ctDNA- vs. RCB-III/ctDNA+ vs. RCB-III/ctDNA- | Persistent negative vs. Cleared at T1 vs.Cleared at T2 vs.Cleared at T3 vs. No clearance at T3 |
| Number of patients | 249 | 249 | 249 | 249 |
| Number of events | 66 | 66 | 66 | 66 |
| ctDNA time points | - | Pretreatment (T0) | Post-NAT (T3) | Pretreatment (T0), Week 3 (T1), Week 12 (T2), Post-NAT (T3) |
| C-index | 0.75 | 0.80 | 0.81 | 0.84 |
| Lower 95% CI | 0.65 | 0.73 | 0.74 | 0.79 |
| Upper 95% CI | 0.84 | 0.87 | 0.87 | 0.90 |

To compare model performance, we analyzed survival data from patients common to all the Cox models (see Fig. 3). CIconfidence interval, NATneoadjuvant therapy, RCBresidual cancer burdenSummary.

having RCB-0/RCB-I after NAT (HR-positive/HER2-negative: Odds ratio (OR) 5.80, 95% CI 2.21–18.30 LR $p < 0.001$; TN: OR 11.50, 95% CI 5.04−28.36, LR $p < 0.001$; HER2-positive: OR 15.0, 95% CI 4.01–73.75, LR $p < 0.001$) (Fig. 4D).

To confirm the association between early ctDNA clearance and favorable response to NAT, we assessed its impact on RCB score distributions. We observed a markedly skewed distribution toward lower RCB scores (favorable response) in patients with early ctDNA clearance at week 3 (T1) in the TN [adjusted Kruskal-Wallis (adj KW), $p < 0.001$] and HER2-positive (adj KW, $p < 0.001$) groups (Fig. 5A). This highly skewed distribution was not observed in the HR-positive/HER2-negative group. However, the median RCB score in patients with early ctDNA clearance at week 3 (T1) was significantly lower than in patients without ctDNA clearance at week 12 (T2, adj KW, $p < 0.001$).

### Predictive value of ctDNA dynamics across treatment types

To examine the predictive value of ctDNA across treatment types, we categorized treatments received by patients with HR-positive/HER2-negative and TN disease into three categories: 1. Paclitaxel (P) alone, 2. P + small molecule inhibitor (SMI), and 3. P + immune checkpoint inhibitor (ICI). The last category consisted of P + HER2-targeted agents administered to patients with HER2-positive disease (Table S3 and Methods).

In patients who received P + ICI (HR-positive/HER2-negative: adj KW $p = 0.013$; TN: adj KW $p = 0.001$), the median RCB score in those with early ctDNA clearance at week 3 (T1) was significantly lower than those with no ctDNA clearance by week 12 (T2) (Fig. 5B, Fig. S7A–B). Similar results were observed in patients receiving HER2-targeted agents (adj KW p < 0.001) (Fig. 5B, Fig. S7C). Moreover, ctDNA clearance at week 3 (T1) in these treatment groups was associated with an increased likelihood of having RCB-0/RCB-I after NAT (Fig. S8). ctDNA clearance at week 3 (T1) was the least predictive of a favorable response in HR-positive/HER2-negative (Fig. S8A) compared to the other subtypes (Fig. S8B-C).

Together, these results demonstrate that ctDNA clearance as early as 3 weeks (T1) after treatment initiation predicts a favorable response to NAT, including in treatments containing ICI and HER2-targeted agents. The predictive value of early ctDNA clearance was stronger in the TN and the HER2-positive subtypes than in the HR-positive/HER2-negative group.

### ctDNA assay variants are conserved in the residual tumors

Of the 387 patients with RCB-II/RCB-III, 316 (82%) had ctDNA data available post-NAT (T3). Intriguingly, only 51 (16%) of these patients were ctDNA-positive despite having a high tumor burden after NAT. We explored whether the ctDNA assay variants initially chosen from the mutation profiling of pretreatment tumors were conserved over time despite the changes in the mutational landscape during NAT. Thus, we checked for the presence of ctDNA assay variants in serial tumor tissues from 94 patients with RCB-II or RCB-III (45 HR-positive/HER2-negative, 38 TN, and 11 HER2-positive). Of the 94, 76 had paired mutation profiling data for tumors from pretreatment (PT0) and week 3 or 12 (on-treatment, PT1'), and 71 had paired mutation profiling data for tumors from pretreatment (PT0) and post-NAT at surgery (PT3) (Fig. 6A).

WES analysis of pretreatment tumors (PT0) in 94 patients with RCB-II or RCB-III detected a median of 603 variants (range 264-1509). There was no significant difference (Kruskal−Wallis $p = 0.4392$) when compared to the median number of variants detected in 76 on-treatment (PT1') tumors (median 588, range 265-6360) and 71 post-NAT (PT3) tumors (median 595, range 277-7468). We then checked for the presence of 16 ctDNA assay variants selected from WES of PT0 in the matched serial tumors. A median of 16 and 15 variants were detected in PT1' and PT3 tumors, respectively.

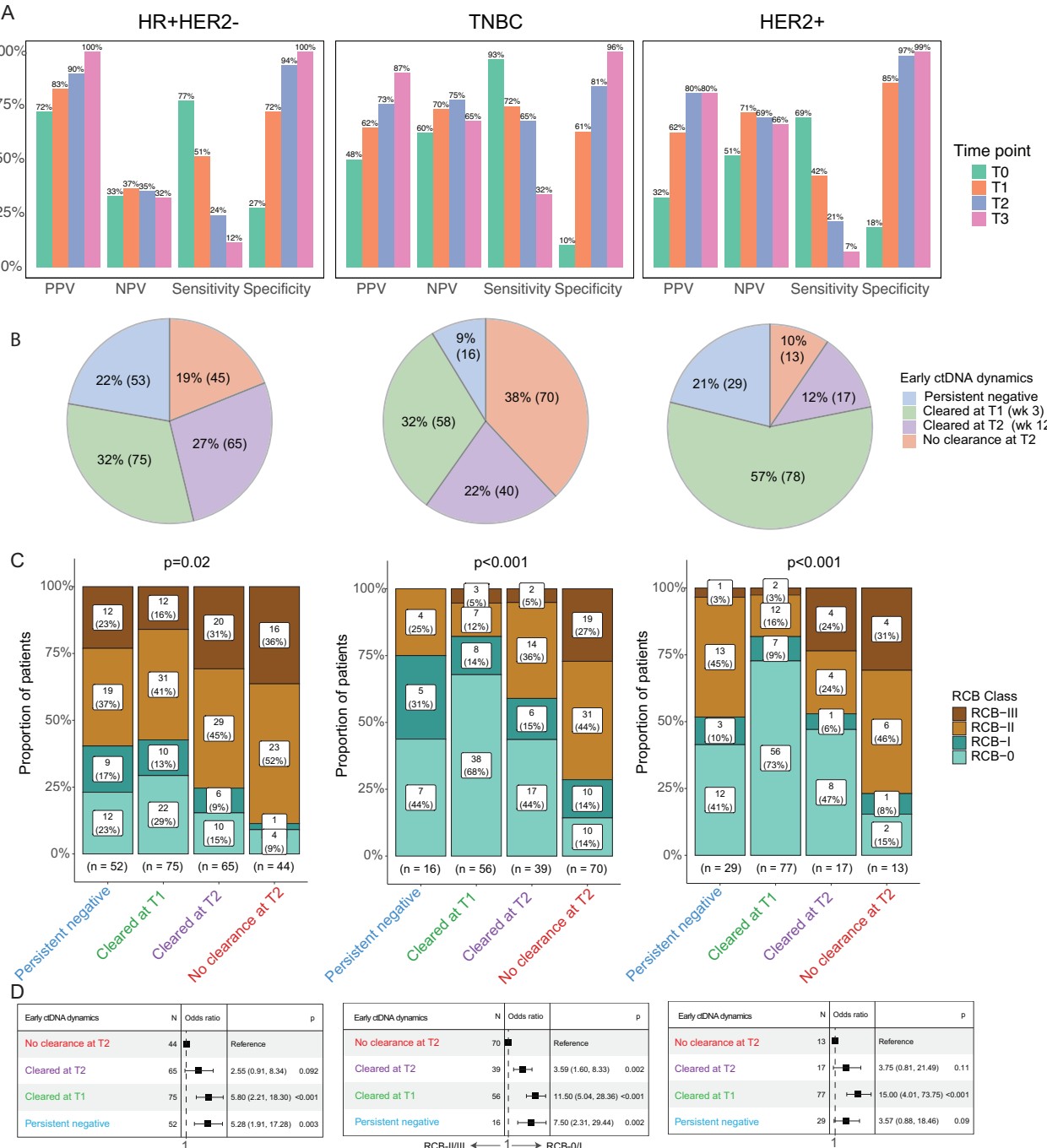

**Fig. 4 | Performance metrics of the ctDNA assay and association between early ctDNA dynamics and. residual cancer burden (RCB) class. A** The positive predictive value (PPV) and sensitivity for predicting moderate or extensive RCB (RCB-II/III), the negative predictive value (NPV) and specificity for predicting pathologic complete response or limited RCB (RCB-0/I) were calculated in patients across receptor subtypes: hormone receptor (HR)-positive/HER2-negative (HR + HER2-, left panel), triple-negative breast cancer (TNBC, middle panel), or HER2-positive (HER2 + , right panel). ctDNA testing was performed at pretreatment (T0), weeks 3 (T1) and 12 (T2) after treatment initiation, and post-NAT before surgery (T3). **B** Pie charts showing the proportion of patients by early ctDNA dynamics groups by subtype. **C** Bar plots showing the proportion of RCB classes in each early ctDNA dynamics group. The percentages may not add up to 100% due to rounding. The *p*-values were calculated using the Chi-squared test. **D** Forest plots from logistic regression analyses showing odds ratio estimates and 95% confidence intervals. The *p*-values were calculated from likelihood ratio tests.

First, we compared the distribution of variant allele frequencies (VAF) of the patient-specific ctDNA variants in the tumor tissue with those in the plasma (ctDNA) at pretreatment (T0, Fig. S9A). We observed significantly higher VAFs in tissue compared to plasma (Wilcoxon *p* < 0.001).

Second, we determined the proportion of all the variants jointly detected by mutation profiling in paired serial tumors (i.e., % conserved).

The comparisons revealed low median conservation rates of 24.8% (range: 4.8–44.9%) between paired PT0 and PT1' tumors and 23.2% (range: 2.8–40.5%) between paired PT0 and PT3 tumors (Fig. S9B). In contrast, when we compared the detection of ctDNA assay variants in paired tumors, we observed high median conservation rates of 96.9% (interquartile range (IQR): 87.5–100%) between paired PT0 and PT1' tumors, and 94.0% (IQR: 83–100%) between paired PT0 and PT3 tumors.

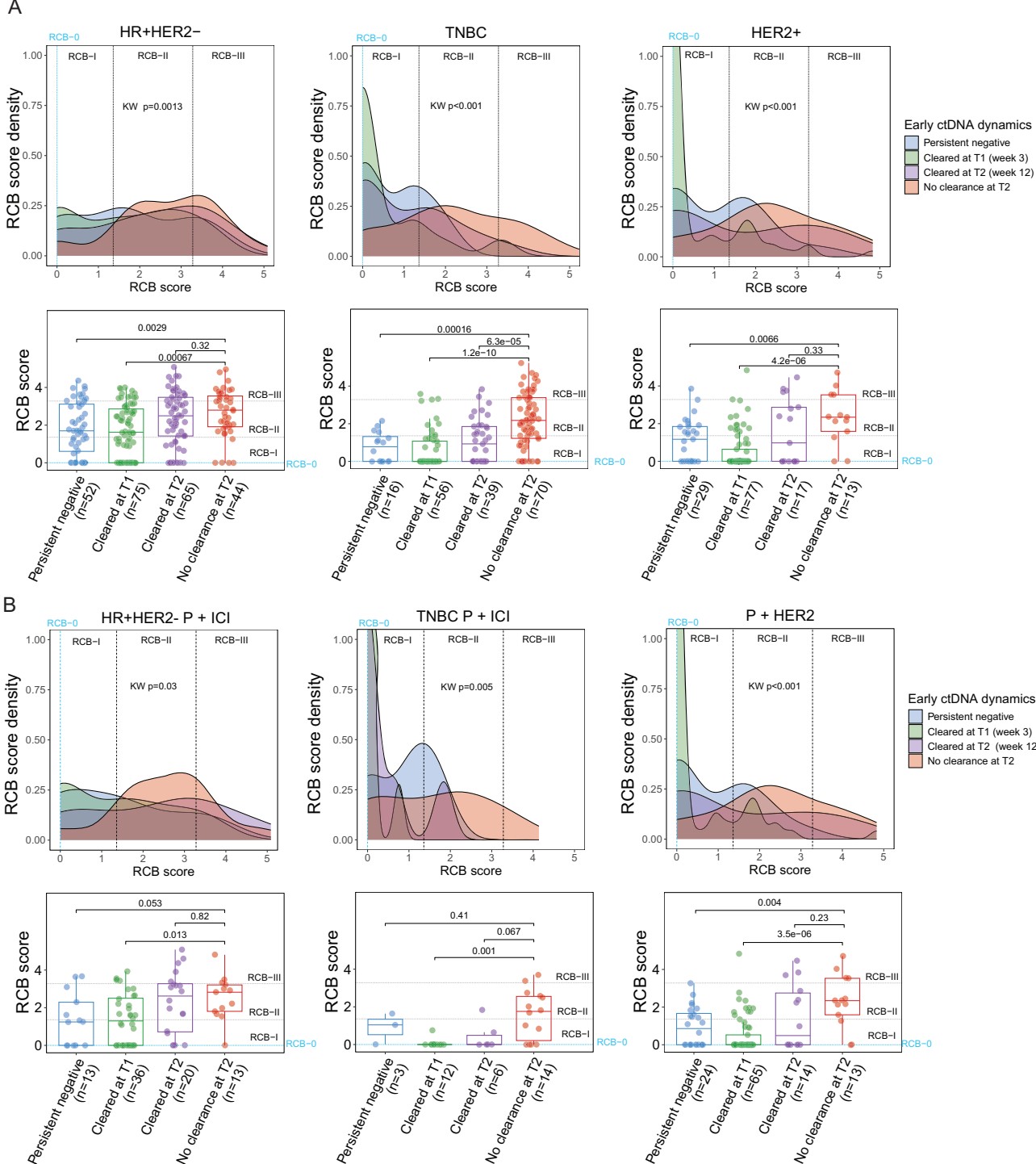

**Fig. 5 | Early ctDNA dynamics predict response and impact the residual cancer burden (RCB) score distribution. A** Patients across receptor subtypes, hormone receptor (HR)-positive/HER2-negative (HR + HER2-), triple-negative breast cancer (TNBC), or HER2-positive (HER2 + ) were stratified by early ctDNA dynamics: persistent ctDNA-negative, ctDNA cleared at week 3 (T1) or week 12 (T2) and no ctDNA clearance at week 12 (T2). The distribution of RCB scores for each group was visualized using density (upper) and box-and-whisker (lower panel) plots. **B** The density (upper) and box-and-whisker (lower panel) plots show the distribution of the RCB scores in patients across receptor subtypes and the type of treatment received stratified by early ctDNA dynamics. The treatment types included paclitaxel combined with immune checkpoint inhibitors (P + ICI) or HER2-targeted

drugs (P + HER2). See Fig. S7 for extended results. The density plots visualize the distribution of continuous values (RCB scores), with peaks showing where the values are concentrated. The total area under each distribution curve is equal to 1. The box-and-whisker plot shows the interquartile range (IQR) of the RCB scores for a given group divided into quartiles, with Q1 (the lower end of the box), Q2 (the median), and Q3 (the upper end of the box). The whiskers from the box represent the data outside the upper and lower quartiles. The *p*-values for multi-group comparisons (> 2) were calculated using the Kruskal-Wallis (KW) and pairwise comparisons using the Wilcoxon rank-sum test with adjustment for multiple hypotheses testing using the Bonferroni correction.

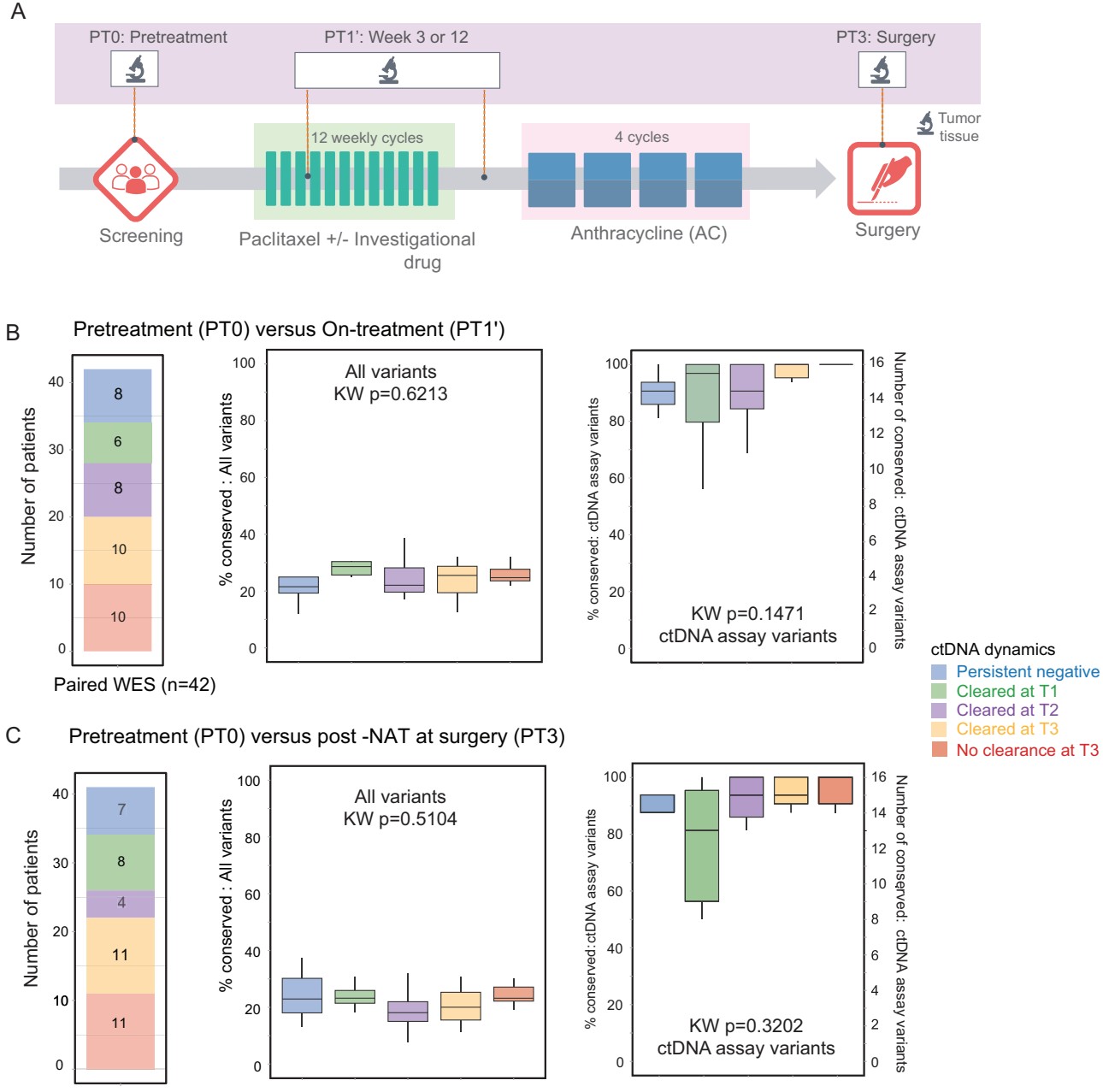

**Fig. 6 | Serial tumor mutation profiling of neoadjuvant therapy (NAT)-resistant tumors reveals the high conservation of patient-specific ctDNA assay variants in the tissue over time. A** Mutation profiling of matched tumor tissue samples was performed at pretreatment (PT0), on-treatment (week 3 or week 12, PT1'), and post-NAT at surgery (PT3) for a subset of patients with RCB-II and RCB-III. Paired mutation profiling data was available for **B** pretreatment (PT0) and on-treatment (PT1') tumors (*n* = 42) and pretreatment (PT0) and post-NAT (PT3) tumors (*n* = 41) in patients with complete ctDNA data for 4 time points. Patients were stratified by ctDNA dynamics: persistent ctDNA-negative, ctDNA cleared at week 3 T1 (T1), week 12 (T2), or post-NAT before surgery (T3) post-NAT before surgery, and no ctDNA clearance post-NAT before surgery. **C** The box plots show the distribution of the percentages of conserved somatic variants detected in paired tumor samples collected at PT0 and PT1' (top panel) and PT0 and PT3 (bottom panel). **C** the distribution of the percentages (left y-axis) or the numbers (right y-axis, up to 16 for each patient) of conserved patient-specific ctDNA assay variants in paired tumor samples collected at PT0 and PT1' (top panel) and PT0 and PT3 (bottom panel). ctDNA testing was performed at pretreatment (T0), 3 weeks after initiation of paclitaxel treatment with or without investigational agents (T1), 12 weeks between paclitaxel-based and anthracycline treatment (T2), and after NAT (T3).

Similar results were observed in a subset analysis involving 53 (56%) patients with serial tumor samples for all three time points (Fig. S9C). Additionally, we found that restricting the tumor analysis to mutations with high VAF (>median VAF) increased the degree of concordance (Fig. S9D). The percentage of variant conserved between PT0 versus PT1' was 24.9% for all variants compared to 29.9% when including only high-VAF variants (Wilcoxon *p* < 0.001). Similar results were observed for PT0 versus PT3 (all variants: 23.3% versus high-VAF variants: 28.9%, Wilcoxon *p* < 0.001). All ctDNA assay variants exhibit high VAFs (> median VAF in the tumor), which is a criterion for inclusion in the panel of 16 patient-specific assays; thus, the restriction does not impact the level of concordance.

Lastly, the same analysis performed in patients grouped by ctDNA dynamics yielded similar results (Fig. 6B, C, Table S4). We observed low

median conservation rates for all variants in paired mutation profiling data from PT0 versus PT1' (range 21.5–28.6%, $n = 42$ pairs, Fig. 6A, B) and PT0 versus PT3 (range 18.1–23.3%, $n = 41$ pairs, Fig. 6B, C). In contrast, ctDNA assay variants were highly conserved (PT0 versus PT1': range 90.6–100%; PT0 versus PT3: range 81.3–100%), irrespective of ctDNA dynamics.

Despite changes in the mutational landscape, we consistently detected ctDNA assay variants in NAT-resistant tumors, even in patients with persistently ctDNA-negative or with ctDNA clearance. Since these tumors have a decreased risk of metastatic recurrence, we posit that low or undetectable ctDNA shedding and ctDNA clearance during treatment in this patient population could be characteristic of NAT-resistant tumors with low metastatic risk.

## Discussion

This study builds on previous research by our group[10–12] and others[13–16,18,25–28] examining the clinical significance of ctDNA in patients with early-stage breast cancer receiving NAT. Our findings demonstrate that ctDNA information can refine risk stratification in patients with tumors resistant to NAT (RCB-II/RCB-III). In these patients who typically have poor prognoses, ctDNA testing fine-tuned risk stratification of NAT-resistant tumors based on their likelihood of metastatic recurrence. ctDNA negativity at pretreatment (T0) and post-NAT (T3) correlated with better survival outcomes and lower risk of metastatic recurrence. Consistent with findings from our previous studies[11,12], we observed that ctDNA dynamics, i.e., persistent ctDNA negativity and early clearance, significantly correlate with improved DRFS, even in patients with RCB-II/III. These results suggest that the lack of ctDNA detection in NAT-resistant tumors is linked to less aggressive biology and a lower propensity for metastasis. In this cohort, the prognostic performance of the RCB class model (as measured by Harrell's concordance index) was consistent with previously reported c-indices[5]. The increase in c-indices observed in mixed models suggests that the prediction of DRFS may be improved when incorporating ctDNA information with RCB.

Understanding the prognostic impact of ctDNA status and the timing of ctDNA clearance[25,26] may inform treatment decisions following surgery in the adjuvant setting[29], offering treatment de-escalation options for patients with RCB-II/RCB-III who are either persistently ctDNA-negative or with early ctDNA clearance during NAT. We found a significant association between ctDNA dynamics and RCB. Early clearance of ctDNA 3 weeks after treatment initiation shifted the distribution toward lower RCB scores (more favorable response), especially in patients with TN and HER2-positive breast cancer. In subgroup analyses by treatment type, early ctDNA clearance (week 3) in the HER2-positive group receiving HER2-targeted agents was associated with a favorable response. A similar association was observed in HR-positive/HER2-negative and TN groups treated with ICI-containing regimens. ctDNA could be an important predictive biomarker that can help maximize the efficacy of immunotherapy[30–32]. Consistent with our previous study in HER2-negative patients[11], the predictive value of ctDNA was less pronounced in the HR-positive/HER2-negative subtype than in the TN group. Moreover, when evaluating the performance of the ctDNA assay for predicting RCB, it is important to consider the tumor's receptor subtype, as the test's performance metrics (such as PPV, NPV, and sensitivity) can vary across these groups.

The ability of ctDNA to predict response early during treatment can aid in therapeutic decisions that could improve response rates to NAT[21,33]. For example, the new I-SPY2.2 trial design aims to maximize the likelihood of achieving an RCB-0 (pathologic complete response) by allowing an early change in therapy in patients predicted to have a poor response to an investigational agent (treatment escalation)[8,9]. In contrast, patients predicted to have RCB-0 can receive surgery early to minimize exposure to the toxicities of unnecessary treatment

(treatment de-escalation). The goal is to combine ctDNA with imaging and pathology assessment, currently used for treatment redirection[8,9], to improve early response prediction to NAT.

Other groups have also investigated the relationship between ctDNA and RCB in smaller cohorts[13–16]. Stecklein and colleagues showed that ctDNA positivity in patients ($N = 80$) with RCB-II or RCB-III predicted worse survival[15]. However, in contrast to our study, which tested ctDNA in the neoadjuvant setting, ctDNA testing was performed 1 to 6 months after all curative treatment. Zhou and colleagues[16] showed that on-treatment persistence of ctDNA ($n = 145$) predicted poor response (RCB-II or RCB-III) to NAT. Parsons et al.[13] ($n = 68$) and Shan et al.[14] ($n = 44$) demonstrated a correlation between ctDNA levels and RCB in TN breast cancers. These studies, however, did not assess the prognostic significance of ctDNA[13,14,16].

Our initial pilot study[12] showed a significantly higher ctDNA positivity rate at pretreatment (T0) in the HER2-positive group versus the HR-positive/HER2-negative group. We did not observe a difference in this larger cohort (77% versus 76%). However, there was a greater decrease in the ctDNA positivity rate in HER2-positive group compared to the HR-positive/HER2-negative group between pretreatment (T0) and 3 weeks after treatment initiation (T1). This is consistent with the higher rates of favorable response to NAT in the HER2-positive group.

Intriguingly, only 16% of resistant tumors (RCB-II and RCB-III) had detectable ctDNA post-NAT before surgery (T3). Since we used a personalized tumor-informed ctDNA test based on patient-specific tumor variants identified in pretreatment tumors (PT0), we assessed whether NAT-mediated selection pressures impacted the presence of these variants in NAT-resistant tumors over time. However, mutational landscape comparisons in solid tumors from the same patients showed that the initial ctDNA assay variants were conserved between on-treatment and post-NAT tumors, even in persistently ctDNA-negative or with ctDNA clearance. These results indicate that changes in the mutational landscapes in NAT-resistant tumors cannot fully account for the lack of ctDNA detection. Since ctDNA-negativity and ctDNA clearance are associated with improved DRFS (discussed above), we speculate that low ctDNA shedding (below the test's limit of detection) during NAT could be an intrinsic biology of NAT-resistant tumors with low metastatic risk.

Comparison of the distribution of VAFs of patient-specific ctDNA assays in tumor tissue versus plasma (ctDNA) yielded results consistent with findings from several studies[34–36]. Differences in VAFs between tumor tissue and plasma (ctDNA) can be due to several technical (e.g., preanalytical conditions and the ctDNA assay used) and biological factors. VAF in the tissue reflects the proportion of altered loci in the tumor, admixed with normal DNA from stromal cells (when present) and tumor subclones that do not carry the alteration. In contrast, VAF in plasma reflects the proportion of tumor-derived molecules (ctDNA) in the background of cell-free DNA[37]. Key contributors to these differences include variable ctDNA shedding, high background of normal cell-free DNA (cfDNA), and tumor status. The rate of ctDNA in plasma is influenced by tumor size, location, vascularization, and cell turnover and death (e.g., apoptosis or necrosis)[38]. In addition, the plasma contains an admixture of cfDNA from both tumor and normal cells (predominantly of hematopoietic origin), which can dilute the concentration of the ctDNA[39]. The concentration of ctDNA in the plasma can also reflect the current biological state of the tumor, which may change depending on treatment response[11,12] and disease status[20].

The study has several limitations. Although the analytic cohort included 723 patients, various analyses were performed in smaller sample sizes. For example, stratification using ctDNA dynamics in patients with RCB-II/RCB-III yielded smaller subsets that precluded multivariable adjustment for confounding effects. We did control for the effects of receptor subtype on DRFS. Additionally, a few patients

who initially tested negative for ctDNA at pretreatment (T0) later tested ctDNA-positive at various subsequent time points. However, due to the limited number of such cases, we did not evaluate the prognostic and predictive implications of these non-monotonic ctDNA fluctuations. The analysis evaluating the predictive value of ctDNA in different treatment types included only the I-SPY2 arms for which ctDNA analysis was completed. Correlative studies using serial mutation profiling data were performed on smaller subsets of patients with available data. Analyzing modest-sized subsets may introduce biases and lead to chance findings; thus, validation studies in larger cohorts are necessary.

Tumor-informed ctDNA tests require tumor tissue for sequencing to identify personalized ctDNA assay variants[20]. Tumor samples may be unavailable, inadequate, or of poor quality, which presents a recognized limitation for this type of test. This could impact clinical implementation, particularly in settings outside of clinical trials, where tumor tissue may be less readily available. However, since the ctDNA assay can be designed based on various sources of tissues, including diagnostic biopsy or surgically resected primary or metastatic tissue[40,41], as well as on-treatment and post-NAT tumors, as suggested by the findings of this study, tumor-informed ctDNA tests are feasible for the majority of patients. Another known limitation of tumor-informed tests, especially if designed using pretreatment tumors, is that they will miss mutations that emerge during or after therapy.

Current efforts in I-SPY2 involve expanding the study to an additional ~700 patients, which could facilitate comparisons of the clinical significance of ctDNA among groups within the HER2-positive (HR-negative/HER2-positive vs. HR-positive/HER2-positive) and histological (invasive lobular carcinoma vs. invasive ductal carcinoma) subtypes. A median follow-up of 4.7 years may be sufficient for the HER2-positive and TN subtypes; however, it may be insufficient for those with HR-positive/HER2-negative disease, where the risk of recurrence persists for decades. Hence, regular follow-up is ongoing in I-SPY2 to monitor for late recurrences in these patients. Lastly, ctDNA detection rates are highly dependent on the selected method and the analytical features of the assay. The results of this study are based on ctDNA detection using a personalized tumor-informed assay (Signatera™ Exome) and are therefore constrained by the performance metrics of this test. Studies involving the retesting of plasma samples to determine whether expanding the panel of patient-specific ctDNA variant assay (e.g., including mutations that emerge in on-treatment and post-NAT tumors or using the Signatera™ Genome test) could improve ctDNA detection rates are being planned.

In summary, we demonstrated that ctDNA dynamics improved the prognostic impact of RCB by refining the risk stratification of NAT-resistant tumors (RCB-II/RCB-III), supporting the potential for ctDNA to complement RCB as an early surrogate for survival. We found that ctDNA clearance, as early as 3 weeks after treatment initiation, predicted response to NAT, including immunotherapy. ctDNA-negativity or ctDNA clearance in NAT-resistant tumors was associated with a lower propensity to metastasize. Refining risk stratification of NAT-resistant tumors into high versus low metastatic risk could guide treatment decisions (de-escalation or escalation) to avoid over-treatment or overcome resistance to NAT.

# Online methods
## Clinical protocol
**Patients.** The study involved patients with high-risk (MammaPrint high) early-stage breast cancer who were enrolled in I-SPY2 (NCT01042379), a multicenter adaptive platform trial for testing therapeutic agents in the neoadjuvant setting (Table S1). The trial design and patient eligibility have been described in detail elsewhere[42]. Institutional Review Boards approved the I-SPY2 protocol at all participating institutions. All patients signed a written informed consent to allow research on their biospecimen samples.

**Blood samples.** Blood was collected at pretreatment (T0), 3 weeks after initiation of paclitaxel treatment with or without an investigational drug (T1), at 12 weeks post-paclitaxel treatment before the anthracycline regimen (T2), and post-NAT before surgery (T3) (Fig. 1).

## Molecular protocol
**Tumor and ctDNA analysis.** A tumor-informed, personalized ctDNA test (Signatera™, Natera Inc.) was used in the study, as previously described[11,12]. Briefly, whole exome sequencing (WES) was performed on formalin-fixed paraffin-embedded (FFPE) tumor tissue and matched-normal DNA from buffy coat to filter germline mutations and variants from clonal hematopoiesis of indeterminate potential. To minimize the impact of low tumor content on variant allele fraction (VAF) interpretation, e.g., subclonal vs. clonal inference[43], only biopsy cores with high tumor cellularity ($\geq 30\%$) were used for WES. Additionally, quality control measures, such as pathology review, were performed to identify the most representative tumor regions for sequencing. Based on the WES results, multiplex PCR primers were designed for 16 tumor-specific, somatic single-nucleotide variants for each patient. A sample was considered ctDNA-positive if $\geq 2$ of 16 variants were detected in the circulating cell-free DNA[11,12].

Less than 50% of on-treatment biopsies contained enough tumor cells for WES. This might be because the biopsy missed the residual tumor or because the targeted clip used to guide the biopsy was in a tumor area responding to treatment.

## Statistical analysis
The statistical tests and data visualization described below were all performed using packages in R.

**Patient population.** The study included 723 patients. Of the 723, 301 were included in previously published studies[11,12]. Data from these patients were combined with data from an additional 422 new patients in this study. The data integration was motivated by a new analysis approach, in which the prognostic value of ctDNA status was assessed within RCB groups (RCB-0/I and RCB-II/III). Combining the data was also essential for evaluating the clinical relevance of ctDNA dynamics within RCB groups and across receptor subtypes, where sample sizes in individual manuscripts[11,12] were limited. Other novel analyses in this report, which leveraged the larger dataset, examined how ctDNA dynamics influence the distribution of RCB scores within receptor subtypes, as well as the predictive value of ctDNA dynamics in patients stratified by the type of treatment received (e.g., ICI-containing regimens).

**Clinicopathologic characteristics.** Participants were grouped by receptor subtypes: hormone receptor (HR)-positive/HER2-negative, triple-negative (TN), and HER2-positive. Baseline clinicopathologic characteristics were compared using Fisher's exact test for categorical variables and the Kruskal-Wallis test for continuous variables.

**Prognostic value of ctDNA.** We assessed whether ctDNA status (ctDNA-positive or ctDNA-negative) at pretreatment (T0) and post-NAT (T3), and ctDNA dynamics can improve the risk stratification of poor-prognosis patients with NAT-resistant tumors (RCB-II and RCB-III).

For ctDNA status, we included only patients who had RCB-II/III with available ctDNA data at both time points ($n = 304$) to ensure that the same patients were involved in both survival analyses, thereby avoiding sampling bias and reducing the impact of confounding variables.

For ctDNA dynamics, patients who had RCB-II/III were classified into 5 groups based on the timing of ctDNA clearance. Only patients with complete ctDNA data for 4 time points were included in the survival analysis. The groupings were defined as:

(1) persistent ctDNA-negative (ctDNA-/-/-/- $n = 45$),
(2) cleared at week 3, T1 (ctDNA + /-/-/- $n = 58$),
(3) cleared at week 12, T2 (ctDNA + /+/-/- $n = 54$),
(4) cleared post-NAT before surgery, T3 (ctDNA + /+/+/- $n = 40$, ctDNA + /-/+/- $n = 11$)
(5) no clearance at T3 (ctDNA + /+/+/+ $n = 29$, ctDNA + /-/+/+ $n = 3$, ctDNA + /+/-/+, $n = 9$).

A total of 8 patients with non-monotonic ctDNA dynamics patterns who tested ctDNA-negative at pretreatment (T0) but tested ctDNA-positive at a later time point (e.g., ctDNA-/-/-/+ $n = 1$, ctDNA-/-/+/- $n = 3$, ctDNA-/+/-/- $n = 4$) were excluded from the survival analysis.

The survival endpoint was distant recurrence-free survival (DRFS), defined as the time from treatment consent to the first distant recurrence or death from any cause. A total of 133 DRFS events were reported (Table S1). Of these, 12 occurred in patients from the original study[12]; 72 events were from patients in the second study, which was limited to the HER2-negative subtype[11]; and 49 additional events involved patients in this study. Patients without events were censored at the last follow-up. We performed the Kaplan-Meier analysis to visualize the survival curves and calculate the 3-year DRFS rates for each group. The $p$-values were calculated using the log-rank test.

To evaluate the prognostic performance of ctDNA, RCB (see next section), and other clinicopathologic variables for predicting DRFS, we fitted Cox proportional hazards models using the coxph function from the survival R package. Univariable (Table S2) and multivariable Cox regression analyses were performed to adjust for potential confounders. Forest plots were generated from Cox regression analyses to visualize the estimates of hazard ratios and their 95% confidence intervals. The $p$-values were calculated using the Wald test.

Model discrimination was assessed using Harrell's concordance index (c-index)[44], a performance metric that evaluates the predictive accuracy of risk models. The c-index quantifies the proportion of usable patient pairs for which the model correctly predicts the order of recurrence events[44]. A c-index of 0.5 corresponds to random chance, and a value of 1.0 reflects perfect concordance between predicted risk and observed outcomes. Linear predictor scores were extracted from the Cox models and used to calculate the c-indices and 95% confidence intervals using the concordance.index function from the survcomp R package[45]. To ensure a fair comparison of model performance, we analyzed survival data from the same patients across the different Cox models. Only patients with ctDNA data at all four time points were included in the analysis ($n = 249$). A total of 66 DRFS events were observed in this subset of patients.

**Predictive value of ctDNA.** Because the pathologic response to NAT varied between receptor subtypes (Table S1)[22], we assessed the predictive value of early ctDNA dynamics across these groups.

Early ctDNA dynamics was defined as the timing of ctDNA clearance from pretreatment (T0) up to 12 weeks (T2). Since the goal of I-SPY2.2 is to use predictive biomarkers to guide therapeutic decisions early during NAT, our predictive models omitted the post-NAT (T3) time point because of its temporal proximity to surgery. Assessment of ctDNA early during NAT (e.g., week 3) will facilitate judicious changes in treatment (escalation/de-escalation) to increase the likelihood of RCB-0 or allow early surgery to minimize exposure to the toxicity of unnecessary treatment.

Only patients with complete ctDNA results from pretreatment (T0) up to 12 weeks (T2) were included in the analysis. The early ctDNA dynamics consisted of 4 groups:
(1) persistent negative (ctDNA-/-/-, $n = 98$),
(2) cleared at week 3, T1 (ctDNA + /-/-, $n = 211$),
(3) cleared at week 12, T2 (ctDNA + /+/-, $n = 122$), and
(4) no clearance at T2 (ctDNA + /+/+, $n = 109$, or ctDNA + /-/-, $n = 19$).

A total of 14 patients who tested ctDNA-negative at pretreatment (T0) but tested ctDNA-positive at a later time point (e.g., ctDNA-/-/+ $n = 3$, ctDNA-/+/+ $n = 2$, ctDNA-/+/- $n = 8$) were excluded from the analysis.

The response endpoints were the RCB score and RCB classes. The RCB score is a continuous measure of the amount of invasive cancer in the breast tumors and regional lymph nodes[5] determined by pathologic examination at the time of surgery. Applying empirically derived cutoffs to the continuous RCB score produced 4 groups: RCB-0, RCB-I, RCB-II, and RCB-III, representing no invasive cancer in the breast and regional lymph nodes (pathologic complete response), limited, moderate, and extensive RCB following NAT, respectively[5]. In logistic regression analyses, we categorized responses into RCB-0/RCB-I versus RCB-II/RCB-III.

We assessed how early ctDNA dynamics affected the distribution of RCB scores using density and box-and-whisker plots. A density plot visualizes the distribution of continuous values (RCB scores), with peaks showing where the values are concentrated. The total area under each distribution curve is equal to 1. A box plot consists of a box representing the interquartile range (IQR) of the RCB scores for a given group divided into quartiles, with Q1 (the lower end of the box), Q2 (the median), and Q3 (the upper end of the box). The whiskers from the box represent the data outside the upper and lower quartiles. The $p$-values for multi-group comparisons (> 2) were calculated using the Kruskal-Wallis test, and the $p$-values for pairwise comparisons using the Wilcoxon rank-sum test with adjustment for multiple hypothesis testing using the Bonferroni correction. The forest plots were generated from logistic regression analyses to visualize odds ratios and 95% confidence intervals. The $p$-values were calculated using the likelihood ratio test.

To assess the predictive performance of the ctDNA assay for predicting RCB, we calculated the positive predictive value (PPV) for predicting moderate or extensive RCB (RCB-II/III), the negative predictive value (NPV) for predicting pathologic complete response or limited RCB (RCB-0/I), and the test's sensitivity and specificity across receptor subtypes and time points during NAT.

**Prediction of response by treatment classification.** The I-SPY2 trial evaluated the efficacy of therapeutic agents combined with a chemotherapy backbone. The control arm consists of weekly paclitaxel (P) for 12 cycles and once every 3 weeks of anthracycline (AC) for 4 cycles (P + AC) (Fig. 1A). Patients with HER2-positive disease received HER2-targeted drugs (Trastuzumab alone or Pertuzumab + Trastuzumab) in addition to P + AC. We also assessed the predictive value of ctDNA by the type of treatment the patients received. This analysis included patients in the control and treatment arms where ctDNA testing was completed. We classified patients into 4 groups by the type of treatment received (Table S3):
(1) patients ($n = 161$) who received P only;
(2) patients ($n = 146$) who received treatment containing small molecule inhibitors (SMI) [Irinotecan + Talazoparib ($n = 49$); ABT 888 + Carboplatin ($n = 58$), MK-2206 ($n = 39$)];
(3) patients ($n = 154$) who received treatment containing immune checkpoint inhibitor (ICI) [anti-PD-1 4 cycles ($n = 46$); anti-PD-1 8 cycles ($n = 56$); Durvalumab + Olaparib ($n = 52$)]; and
(4) patients ($n = 161$) who received HER2-targeted drugs [Pertuzumab + Trastuzumab ($n = 111$), Trastuzumab ($n = 8$), T-DM1 + Pertuzumab ($n = 42$)].

**Tumor mutation profiling.** Serial mutation tissue profiling data ($n = 94$) from a subset of patients with RCB-II/RCB-III were available for analysis. Tissues collected at pretreatment needle biopsy, on-treatment needle biopsy (week 3 or week 12 after treatment initiation), and post-NAT at surgery were denoted as PT0, PT1', and PT3,

respectively. Tumor tissue mutation profiling was performed as previously described[11]. Only tumor tissue samples with a tumor purity greater than 20% were included in the analysis. Tumor purity was calculated from alignment bam files and the R package TitanCNA (v1.44.0). Variants in the vcf files generated from the mutation profiling for each serial tumor tissue were used for the variant conservation analysis. Variants with an allele frequency lower than 1% were excluded from the analysis. A variant was deemed "conserved" if it was detected in matched serial tumor samples, i.e., shared variants. The percentage of variant conservation was calculated as the number of shared variants between paired tissue samples [pretreatment (PT0) versus on-treatment (PT1′) tumors or pretreatment (PT0) versus post-NAT (PT3) tumors] divided by the number of variants detected in pretreatment (PT0) tumors.

For ctDNA dynamic pattern analysis, only patients with the complete set of ctDNA data for 4 time points [pretreatment (T0), week 3 (T1), week 12 (T2), and post-NAT (T3)] were included, $n = 42$ and $n = 41$ for PT0 versus PT1′ and PT0 versus PT3, respectively. Patients were divided into 5 groups based on ctDNA dynamics: persistently negative, ctDNA-positive at T0 but cleared at week 3 (T1), week 12 (T2), and post-NAT (T3), or no clearance post-NAT (T3). The ANOVA or Kruskal-Wallis tests were used to test significant differences between variant conservation across paired tissue samples and ctDNA dynamics.

### Reporting summary

Further information on research design is available in the Nature Portfolio Reporting Summary linked to this article.

## Data availability

De-identified subject-level data are made available to academic and not-for-profit investigators for projects approved by the I-SPY Data Access and Publications Committee. Details of the application and review process are available at: https://www.quantumleaphealth.org/for-investigators/clinicians-proposal-submissions/. Data is made available to approved applicants within 1 to 2 months of application.

## Code availability

This study used only publicly available software; no custom code was generated.

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

## Acknowledgements

This research was supported by the National Cancer Institute (NCI) of the National Institutes of Health (NIH) under award R01CA255442. Additional funding was provided by NCI NIH award number P01CA210961. The authors wish to acknowledge the generous support of the study sponsors, Quantum Leap Healthcare Collaborative (QLHC, 2013 to present), and the Foundation for the National Institutes of Health (2010 to 2012). The authors sincerely appreciate the ongoing support for the I-SPY2 Trial from the Safeway Foundation, the William K. Bowes, Jr. Foundation, Give Breast Cancer the Boot, and the Breast Cancer Research Foundation. Sincere thanks to all the patients who have volunteered to participate in I-SPY2. This study is in collaboration with Merck Sharp & Dohme LLC, a subsidiary of Merck & Co., Inc., Rahway, NJ, USA

## Author contributions

Study design: M.J.M.M., C.Y., L.J.v.V., and L.J.E. Formal analysis: M.J.M.M., N.A.M., Z.A., C.Y., and D.M.W. Enrolled patients: A.D., A.S.C., A.Z., C.I., H.S.R., L.J.E., P.R.P., and R.A.S. Advocacy: A.L.D. Lab studies: L.B.-S., G.L.H., M.J.M.M., and L.J.v.V. Administrative: G.L.H. First draft: M.J.M.M., N.A.M., C.Y., and D.M.W. Editing and review: A.D., A.L.D., A.R., A.S.C., A.T., A.Z., C.I., C.Y., D.M.W., D.R., E.K., G.L.H., H.S.R., L.B.-S., L.J.E., L.J.v.V., M.C.L., M.J.M.M., N.A.M., P.R.P., R.A.S., R.W.S., S.R.H., W.L., and Z.A. Leadership: M.J.M.M., C.Y., H.S.R., A.D, L.J.v.V., and L.J.E.

## Competing interests

S.R.H., A.T., D.R., E.K., A.R. and M.C.L. are employees of Natera. GLH reports institutional research grant from NIH (1R01CA255442). C.Y. reports institutional research grant from NCI/NIH; salary support and travel reimbursement from Quantum Leap Healthcare Collaborative; US patent titled, "Breast cancer response prediction subtypes," (No. 18/174,491); and University of California Inventor Share. C.I. reports institutional research funding from Tesaro/GSK, Seattle Genetics, Pfizer, AstraZeneca, BMS, Genentech and Novartis; consultancy roles with Arvinas, AstraZeneca, Genentech, Novartis, Pfizer, Gilead, Merck and Seattle Genetics; and royalties from Wolters Kluwer and McGraw Hill. R.A.S. reports institutional research funding from OBI Pharma, Quantum Leap Healthcare, AstraZeneca and Gilead; serves on AstraZeneca and Stemline advisory boards and Gilead speaker's bureau; and has a consultancy role with Quantum Leap Healthcare. A.S.C. reports institutional research funding from Novartis and Lilly. A.Z. reports institutional research funding from Merck; honoraria for Medscape; participation on Pfizer Advisory Board. P.R.P. reports institutional research funding from Genentech/Roche, Fabre-Kramer, Advanced Cancer Therapeutics, Caris Centers of Excellence, Pfizer, Pieris Pharmaceuticals, Cascadian Therapeutics, BOLT, Byondis, Seagen, Orum Therapeutics, Carisma Therapeutics; consulting fees from Personalized Cancer Therapy, OncoPlex Diagnostics, Immunonet BioSciences, Pfizer, HERON, Puma Biotechnology, Sirtex, CARIS Lifesciences, Juniper, Bolt Biotherapeutics, Abbvie; honoraria from Dava Oncology, OncLive/MJH Life Sciences, Frontiers - Publisher, SABCS, ASCO; Speakers' Bureau: Genentech/Roche (past); patents United States Patent no. 8,486,413, United States Patent no. 8,501,417, United States Patent no. 9,023,362, United States Patent no. 9,745,377; uncompensated roles with Pfizer, Seagen and Jazz. L.J.E. reports funding from Merck & Co.; participation on an advisory board for Blue Cross Blue Shield; and personal fees from UpToDate; unpaid board member of Quantum Leap Healthcare. H.S.R. reports institutional research support from AstraZeneca, Daiichi Sankyo, Inc., F. Hoffmann-La Roche AG/Genentech, Inc., Gilead Sciences, Inc., Lilly; Merck & Co., Novartis Pharmaceuticals Corporation, Pfizer, Stemline Therapeutics, OBI Pharma, Ambrx, Greenwich Pharma; advisory and consulting roles with Chugai, Puma, Sanofi, Napo, and Helsinn. A.D.M. reports institutional research funding from Novartis, Pfizer, Genentech and Neogenomics; Program Chair, Scientific Advisory Committee, ASCO. L.J.v.V. is an advisor and shareholder of Exai Bio; part-time employee and owns stock in Agendia. All other authors declare no competing interests.
