## [Transparent Peer Review file · Nature Communications]

Circulating tumor DNA refines risk stratification of neoadjuvant therapy-resistant breast tumors

Corresponding Author: Dr Mark Jesus Magbanua

Version 0:

Reviewer comments:

Reviewer #1

(Remarks to the Author)

In this study, "Circulating tumor DNA refines risk stratification of neoadjuvant therapy-resistant breast tumors" the authors present an updated analysis of ctDNA dynamics from the I-SPY clinical trial, including all three breast cancer receptor subtypes and evaluate the associations between on treatment ctDNA dynamics and the association with recurrence outcomes. This analysis represents the most comprehensive update of this data to date, building upon prior reports in the *Annals of Oncology* (2021) and *Cancer Cell* (2023). Notably, it includes an increase in the number of patients with evaluable ctDNA data, particularly among those with HER2-positive disease who were not included in the most recent *Cancer Cell* publication. While the expanded dataset strengthens the findings, the results are largely consistent with their prior literature and that reported by other groups: a higher amount of residual disease (measured by RCB) at surgery is prognostic, and persistence of ctDNA throughout neoadjuvant therapy further refines risk stratification.

In general, the manuscript is well written, the analyses are appropriate, and the conclusions are consistent with the results. However, there are several areas of clarification which would further strengthen this work.

General points:

-“ctDNA detection” is dependent on the analytical properties of the assay used. Therefore all of the analyses essentially represent “ctDNA detection with the assay used (eg. signatera exome)”. Those properties (eg. limit of detection), absolute values of ctDNA detected, and the consequences of this need to be clear to readers throughout. This will permit appropriate contextualization of these findings in the current study and for other assays that may be more (higher detection at baseline, less clearance on treatment timepoints) or less (lower detection at baseline, more clearance on treatment) sensitive than the one used.

-Can the authors clarify whether all patients included in the current analysis were also part of the previously published articles? Additionally, please confirm whether any panels or timepoints were reanalyzed across the three manuscripts, and if so, specify which ones and provide a rationale for the reanalysis.

-From Fig. 1B, it appears as though not all patients had all 4 timepoints collected. For the analysis of prognostic value, only patients with T0 and T3 timepoints were included, however the authors divided the patients into 5 groups based on the clearance at defined timepoints (line 296-300). Further clarification on how the authors handled patients with missing data in these analyses would be important to clarify in their methods (ie. T0: positive, T1: positive, T2: missing, T3: negative). This approach would also exclude patients who became ctDNA positive on treatment, if it occurred. Discussion if this occurred and a Sup. Table with all patients and their results at each timepoint (consistent with their previous publications) would further improve the clarity of the manuscript.

-Calculating the C-index in this manuscript, given the number of patients, timepoints, and events, would substantially strengthen the analysis and support the utility of ctDNA monitoring as an independent prognostic measure in this setting.

-While the prognostic and predictive value were assessed, the authors did not include measures such as sensitivity, specificity, PPV and NPV of their chosen ctDNA timepoints. This is especially important with the design of the iSPY2.2 trial because although prognostic, the expected utility in refining patient selection is not defined in the current manuscript.

-The authors report a general consistency between variants detected/selected in tumor WES and ctDNA panels for a subset of patients with on-treatment and post-NAT tumor sequencing. While potentially beyond the scope of this manuscript, if additional plasma samples were available for cases where variants differed substantially from baseline to later timepoints, it would be of interest to assess whether ctDNA detection improved over time for these cases with monitoring the post-treatment variants. Additional clarification on whether any patients had samples included at all three timepoints (PT0, PT1',

and PT3), and whether differences were observed between ctDNA panels and tumor variants across these timepoints, would be a valuable addition to the supplemental data.

Minor points:

- Table S1 should include additional information such as: median follow up (and range), number of recurrences per receptor subgroup, and a breakdown of Grade 1 vs. Grade 2 (instead of grouping them together).
- While a median follow-up of 4.7 years is sufficient in patients with HER2+ and TNBC, it is limited for those with HR+/HER2- disease where recurrence risk persists for decades. This should be acknowledged in the discussion.
- The authors should explain why mutation data was not available for all profiled patients with on treatment biopsies (Like 177-178).
- The variables in Figure S1B are not named consistently with Figure S1A
- Discussion or inclusion of the number of DRFS events which have occurred in patients included in the Annals of Oncology and Cancer Cell manuscripts would be informative

Reviewer #2

(Remarks to the Author)

Reviewer #3

(Remarks to the Author)

In the manuscript titled "Circulating tumor DNA refines risk stratification of neoadjuvant therapy-resistant breast tumors," Magbanua and colleagues investigate whether a tissue-informed circulating tumor DNA (ctDNA) assay can enhance risk stratification in patients with breast tumors resistant to neoadjuvant therapy (NAT) and predict treatment response. The study includes 723 patients with all breast cancer subtypes enrolled in the I-SPY2 trial. Among the evaluable patients, 300 (41%) had HR-positive/HER2-negative tumors, 237 (33%) had triple-negative (TN) tumors, and 186 (26%) had HER2-positive tumors. A tissue-informed ctDNA assay was applied at four time points during therapy. Key findings include: (1) ctDNA negativity was associated with significantly improved 3-year distant recurrence-free survival (DRFS) in patients with residual disease (RCB-II and III), and (2) ctDNA clearance by week 3 of neoadjuvant therapy predicted a favorable treatment response. In a subset of 94 residual tumors (45 HR-positive/HER2-negative, 38 TN, and 11 HER2-positive), the authors also assessed mutation conservation across serial tumor biopsies. They found a mutation conservation rate of <30% across tumor tissue samples compared to >90% in serial ctDNA samples. These findings suggest that low or undetectable ctDNA shedding and early ctDNA clearance may be characteristic of certain NAT-resistant tumors with inherently low metastatic potential. The manuscript is well written and addresses important, clinically relevant questions. I believe it would be of interest to your journal's readership. However, I recommend the following revisions:

Major

- In the section "ctDNA improves risk stratification of NAT-resistant tumors," please include outcomes stratified by subtype, as differences in biology and ctDNA kinetics by subtype may yield important insights. If the numbers are too small to support meaningful conclusions, please clarify this in the text.
- Line 142: The sentence, "The proportion of patients who eventually achieved an RCB-0/RCB-I was the highest in this group in all receptor subtypes [HR-positive/HER2-negative: 42%, Chi-squared (χ^2) $p=0.02$]" — please verify whether this statement is accurate for the HR+/HER2- group. As written, 58% of these patients did not achieve RCB 0/I, which raises the question of whether this group indeed had the highest proportion.
- Consider adding if there were differences in ctDNA dynamics by tumor size, nodal status
- Limitations: describe limitations of Signatera, including the sensitivity, describe if any tumor informed assays could not be performed due to the lack of tissue, and how this could impact clinical implementation (as patients are more likely to lack available tissue if they are not enrolled in a clinical trial).
- Supplementary table 2: please add more detail about the treatments received

Minor

- In the introduction, please clarify that pathologic complete response (pCR) is a strong predictor of long-term outcomes in HER2-positive and triple-negative breast cancer, but that its prognostic value is less robust in HR-positive/HER2-negative disease.
- Consider adding background in the introduction about the median conservation rate of mutations across paired tumor samples and its potential clinical implications.
- Line 158: Please spell out "ICI" (immune checkpoint inhibitor) at first mention for clarity.
- Figure 2B: consider adding graphs by subtype
- Figures 3B and 3C: ctDNA- appears to have a strongest correlations with outcomes vs RCBII vs III, this is very interesting and could be included in the text
- If available, it may be interesting to add data on HER2+/HR+ vs HER2+/HR-, there is minimal literature in this setting and this could strengthen the paper

Reviewer #4

(Remarks to the Author)

The authors present data from a large and well-designed longitudinal study of circulating tumor DNA (ctDNA) in early stage breast cancer patients undergoing neoadjuvant therapy. The study includes a substantial cohort (>700 patients) with serial sampling at four timepoints, and explores associations between ctDNA dynamics, pathological response, and recurrence-free survival. The methodology is rigorous and clearly described. The results convincingly demonstrate the potential of serial ctDNA profiling as a clinically informative tool in this setting. The only major limitation is the final analysis concerning ctDNA assay variant conservation over time, which would benefit from further clarification and refinement (see Major Point 1 below).

Major Points

1. Analysis of ctDNA Assay Variant Conservation (Figure 6)

The manuscript reports low median conservation rates of 24.8% (range: 4.8–44.9%) between paired PT0 and PT1' tumors (Figure S5A), and 23.2% (range: 2.8–40.5%) between PT0 and PT3 tumors (Figure S5B), suggesting changes in the tumor mutational landscape.

However, the low concordance between tumor samples at different timepoints may reflect sampling variability and tumor heterogeneity—particularly the presence of spatially distributed subclones—rather than true biological shifts in the mutational landscape. The authors should clarify the specific question this analysis aims to address. If the goal is to evaluate variant conservation between tumor and ctDNA assays, additional context is needed.

Specifically, the authors should:

- * Define what they mean by "variant conservation" and how it is calculated.
 - * Compare the total number of variants detected in tumor vs. ctDNA assays.
 - * Examine the distribution of variant allele frequencies (VAFs) in both tumor and plasma samples.
 - * Assess whether restricting the tumor analysis to high-VAF (likely clonal) mutations alters the degree of concordance.
- These additions would strengthen the interpretation and provide greater insight into the biological and technical factors influencing variant detection over time.

2. Multivariate Statistical Analysis

The study would benefit from a multivariate analysis incorporating ctDNA status, tumor stage, RCB, and other available clinical prognostic factors. This would allow a more robust evaluation of the independent prognostic value of ctDNA and facilitate comparison with established clinical predictors.

Minor Point

The authors are encouraged to explore and report any differences or similarities in ctDNA dynamics and predictive value between ductal and lobular breast cancer subtypes within their cohort, if data allow. This could provide useful insight into subtype-specific applicability of ctDNA monitoring.

Version 1:

Reviewer comments:

Reviewer #1

(Remarks to the Author)

The authors have provided a comprehensive response to the initial reviews. The manuscript has been substantially improved. A few minor points remain:

Consider incorporating Figure S5 into the main manuscript (e.g., as part of Figure 3). Doing so would not only highlight the prognostic value of the findings but also clearly present the associated performance metrics, which would enhance the impact of the results.

Language that implies judgement on the patient, ie patient "achieved" (or conversely didn't achieve), or "patient cleared" should be avoided. Instead, neutral language that captures the treatment effect should be used. Eg. pCR was observed, ctDNA clearance occurred or was observed, or the treatment resulted in ctDNA clearance etc

The authors note that additional patients are currently being evaluated as part of an expanded analysis. Given reported whole-genome based Signatera Assays, the authors should be encouraged to compare the clinical performance of these methods.

Reviewer #2

(Remarks to the Author)

Reviewer #3

(Remarks to the Author)

In "Circulating tumor DNA refines risk stratification of neoadjuvant therapy-resistant breast tumors," Magbanua and colleagues evaluate a tissue-informed circulating tumor DNA (ctDNA) assay in 723 I-SPY2 participants with breast tumors resistant to neoadjuvant therapy (NAT). Patients included HR-positive/HER2-negative (41%), triple-negative (33%), and HER2-positive (26%) subtypes. ctDNA was measured at four time points. ctDNA negativity was linked to improved 3-year distant recurrence-free survival in patients with residual disease, and clearance by week 3 predicted favorable response. In 94 residual tumors, mutation conservation was <30% in tissue but >90% in serial ctDNA, suggesting ctDNA better reflects tumor evolution. These findings indicate that low/undetectable ctDNA and early clearance may characterize NAT-resistant tumors with low metastatic potential. The manuscript is well written and clinically relevant. T

The authors did an excellent job addressing my prior comments. The paper is now clearer, and with the additional information about subtypes, readers can better understand ctDNA dynamics.

My only minor editorial comment is regarding Figure S3B: the time points shown above the subtypes may be confusing since they overlap with the x-axis. Shifting them slightly could improve clarity.

Reviewer #4

(Remarks to the Author)

The authors have addressed all my concerns, thanks!

One minor comment, Table S2 appears to have a typo ("RCB I" repeated twice).

The authors thank the Reviewers for the insightful critique. Below, in blue font, are our detailed responses to each point.

Reviewer #1 (Remarks to the Author):

In this study, "Circulating tumor DNA refines risk stratification of neoadjuvant therapy-resistant breast tumors" the authors present an updated analysis of ctDNA dynamics from the I-SPY clinical trial, including all three breast cancer receptor subtypes and evaluate the associations between on treatment ctDNA dynamics and the association with recurrence outcomes. This analysis represents the most comprehensive update of this data to date, building upon prior reports in the Annals of Oncology (2021) and Cancer Cell (2023). Notably, it includes an increase in the number of patients with evaluable ctDNA data, particularly among those with HER2-positive disease who were not included in the most recent Cancer Cell publication. While the expanded dataset strengthens the findings, the results are largely consistent with their prior literature and that reported by other groups: a higher amount of residual disease (measured by RCB) at surgery is prognostic, and persistence of ctDNA throughout neoadjuvant therapy further refines risk stratification.

In general, the manuscript is well written, the analyses are appropriate, and the conclusions are consistent with the results. However, there are several areas of clarification which would further strengthen this work.

We thank Reviewer #1 for the insightful and helpful comments.

General points:

-“ctDNA detection” is dependent on the analytical properties of the assay used. Therefore all of the analyses essentially represent “ctDNA detection with the assay used (eg. signatera exome)”. Those properties (eg. limit of detection), absolute values of ctDNA detected, and the consequences of this need to be clear to readers throughout. This will permit appropriate contextualization of these findings in the current study and for other assays that may be more (higher detection at baseline, less clearance on treatment timepoints) or less (lower detection at baseline, more clearance on treatment) sensitive than the one used.

RESPONSE: We have also revised the **Discussion** to provide context for the study's findings.

ctDNA detection rates are highly dependent on the selected method and the analytical features of the assay. The results of this study are based on ctDNA detection using a personalized tumor-informed assay (Signatera™) and are therefore constrained by the performance metrics of this test.

-Can the authors clarify whether all patients included in the current analysis were also part of the previously published articles? Additionally, please confirm whether any panels or timepoints were reanalyzed across the three manuscripts, and if so, specify which ones and provide a rationale for the reanalysis.

RESPONSE: No, only 301 of the 723 patients were included in the previously published articles. This study includes 422 new patients along with their ctDNA results. We have revised the **Methods** as follows:

Of the 723, 301 were included in previously published studies^{1,2}. Data from these patients were combined with data from an additional 422 new patients in this study. The data integration was motivated by a new analysis approach, in which the prognostic value of ctDNA status was assessed within RCB groups (RCB-0/I and RCB-II/III). Combining the data was also essential for evaluating the clinical relevance of ctDNA dynamics within RCB groups and across receptor subtypes, where sample sizes in individual manuscripts^{1,2} were limited. Other novel analyses in this report, which leveraged the larger dataset, examined how ctDNA dynamics influence the distribution of RCB scores within receptor subtypes, as well as the predictive value of ctDNA dynamics in patients stratified by the type of treatment received (e.g., ICI-containing regimens).

-From Fig. 1B, it appears as though not all patients had all 4 timepoints collected. For the analysis of prognostic value, only patients with T0 and T3 timepoints were included, however the authors divided the patients into 5 groups based on the clearance at defined timepoints (line 296-300). Further clarification on how the authors handled patients with missing data in these analyses would be important to clarify in their methods (ie. T0: positive, T1: positive, T2: missing, T3: negative). This approach would also exclude patients who became ctDNA positive on treatment, if it occurred. Discussion if this occurred and a Sup. Table with all

patients and their results at each timepoint (consistent with their previous publications) would further improve the clarity of the manuscript.

RESPONSE: We have revised the manuscript to clarify these points as follows:

Methods

We assessed whether ctDNA status (ctDNA-positive or ctDNA-negative) at pretreatment (T0) and post-NAT (T3), and ctDNA dynamics can improve the risk stratification of poor-prognosis patients with NAT-resistant tumors (RCB-II and RCB-III).

For ctDNA status, we included only patients who had RCB-II/III with available ctDNA data at both time points (n=304) to ensure that the same patients were involved in both survival analyses, thereby avoiding sampling bias and reducing the impact of confounding variables.

For ctDNA dynamics, patients who had RCB-II/III were classified into 5 groups based on the timing of ctDNA clearance. Only patients with complete ctDNA data for 4 time points were included in the survival analysis. The groupings were defined as:

- (1) persistent ctDNA-negative (ctDNA-/-/-/- n=45),
- (2) cleared at week 3, T1 (ctDNA+/-/-/- n=58),
- (3) cleared at week 12, T2 (ctDNA+/-/-/- n=54),
- (4) cleared post-NAT before surgery, T3 (ctDNA+/-/-/- n=40, ctDNA+/-/-/- n=11)
- (5) no clearance at T3 (ctDNA+/-/-/- n=29, ctDNA+/-/-/- n=3, ctDNA+/-/-/- n=9).

A total of 8 patients with non-monotonic ctDNA dynamics patterns who tested ctDNA-negative at pretreatment (T0) but tested ctDNA-positive at a later time point (e.g., ctDNA-/-/-/+ n=1, ctDNA-/-/-/+ n=3, ctDNA-/-/-/+ n=4) were excluded from the survival analysis.

Discussion

A few patients who initially tested negative for ctDNA at pretreatment (T0) later tested ctDNA-positive at various subsequent time points. However, due to the limited number of such cases, we did not evaluate the prognostic and predictive implications of these non-monotonic ctDNA fluctuations.

Data availability.

The new I-SPY2 policy no longer permits data sharing as Supplementary Information. Instead, we have added a Data Availability statement to explain how the data can be requested.

De-identified subject-level data are made available to academic and not-for-profit investigators for projects approved by the I-SPY Data Access and Publications Committee. Details of the application and review process are available at: <https://www.quantumleaphealth.org/for-investigators/clinicians-proposal-submissions/>. Data is made available to approved applicants within 1 to 2 months of application.

-Calculating the C-index in this manuscript, given the number of patients, timepoints, and events, would substantially strengthen the analysis and support the utility of ctDNA monitoring as an independent prognostic measure in this setting.

RESPONSE: We have revised **Figure 3**, adding a panel showing c-indices, the number of patients, time points, and events for each of the Cox models in Figures 3A-D. We have also revised the manuscript as follows:

Methods

Model discrimination was assessed using Harrell's concordance index (c-index)³, a performance metric that evaluates the predictive accuracy of risk models. The c-index quantifies the proportion of usable patient pairs for which the model correctly predicts the order of recurrence events³. A c-index of 0.5 corresponds to random chance, and a value of 1.0 reflects perfect concordance between predicted risk and observed outcomes. Linear predictor scores were extracted from the Cox models and used to calculate the c-indices and 95% confidence intervals using the concordance.index function from the survcomp R package⁴. To ensure a fair comparison of

model performance, we analyzed survival data from the same patients across the different Cox models. Only patients with ctDNA data at all four timepoints were included in the analysis (n=249). A total of 66 DRFS events were observed in this subset of patients.

Results

We assessed the prognostic performance of the survival (Cox) models for predicting DRFS. To compare model performance, we analyzed survival data from patients common to all Cox models (n=249). Evaluation of the prognostic performance of the RCB class model (RCB-II vs. RCB-III in **Figure 3A**) revealed a Harrell's c-index of 0.75 (see **Methods, Figure 3E**). Numerically higher c-indices were observed in models that include ctDNA information with RCB-II/III as predictors of DRFS (**Figures 3B-3D**).

Figure 3E. Performance of prognostic models. Model discrimination was assessed using the concordance index (c-index) to evaluate the prognostic performance of the risk models for predicting DRFS. The error bars represent the 95% confidence intervals (CI).

Discussion

In this cohort, the prognostic performance of the RCB class model (as measured by Harrell's concordance index) was consistent with previously reported c-indices⁵. The increase in c-indices observed in mixed models suggests that the prediction of DRFS may be improved when incorporating ctDNA information with RCB.

-While the prognostic and predictive value were assessed, the authors did not include measures such as sensitivity, specificity, PPV and NPV of their chosen ctDNA timepoints. This is especially important with the design of the iSPY2.2 trial because although prognostic, the expected utility in refining patient selection is not defined in the current manuscript.

RESPONSE: We have added **Supplementary Figure 5** to show the performance metrics of ctDNA across all time points in all patients and within receptor subtypes for predicting residual cancer burden

We have revised the manuscript as follows:

Methods

To assess the predictive performance of the ctDNA assay for predicting RCB, we calculated the test's (1) positive predictive value (PPV) for predicting moderate or extensive RCB (RCB-II/III), (2) the negative predictive value

(NPV) for predicting pathologic complete response or limited RCB (RCB-0/I), as well as its (3) sensitivity and (4) specificity across receptor subtypes and time points during NAT.

Results

First, we assessed the predictive performance of the ctDNA assay by measuring the test's positive predictive value (PPV) and sensitivity in correctly identifying patients with RCB-II/III, as well as the negative predictive value (NPV) and specificity in correctly identifying patients with RCB-0/I.

The analysis revealed increasing PPV across time points for all receptor subtypes, but not for NPV (Figure S5). Higher PPV and lower NPV at all time points were observed in the HR-positive/HER2-negative subtype compared to the other receptor subtypes. Furthermore, the sensitivity of the test diminished over time across all receptor subtypes, whereas its specificity improved. Sensitivity was highest in the TN group, while specificity was comparable across all receptor subtypes.

Discussion

When evaluating the performance of the ctDNA assay for predicting RCB, it is important to consider the tumor's receptor subtype, as the test's performance metrics (such as PPV, NPV, and sensitivity) can vary across these groups.

Figure S5. Performance metrics of the ctDNA assay for predicting residual cancer burden (RCB) status (RCB-0/I versus RCB-II/III) across receptor subtypes and time points during neoadjuvant therapy (NAT). The positive predictive value (PPV) and sensitivity for predicting moderate or extensive RCB (RCB-II/III), the negative predictive value (NPV) and specificity for predicting pathologic complete response or limited RCB (RCB-0/I) were calculated in all patients and patients stratified by receptor subtype: hormone receptor (HR)-positive/HER2-negative (HR+HER2-), triple-negative breast cancer (TNBC), and HER2-positive (HER2+). ctDNA testing was performed at pretreatment (T0), weeks 3 (T1) and 12 (T2) after treatment initiation, and post-NAT before surgery (T3).

-The authors report a general consistency between variants detected/selected in tumor WES and ctDNA panels for a subset of patients with on-treatment and post-NAT tumor sequencing. While potentially beyond the scope of this manuscript, if additional plasma samples were available for cases where variants differed substantially from baseline to later timepoints, it would be of interest to assess whether ctDNA detection improved over time for these cases with monitoring the post-treatment variants.

RESPONSE: We agree. We have revised the Discussion as follows:

Studies involving the retesting of plasma samples to determine whether expanding the panel of patient-specific ctDNA variant assays—to include mutations that emerge in on-treatment and post-NAT tumors—could improve ctDNA detection rates are being planned.

Additional clarification on whether any patients had samples included at all three timepoints (PT0, PT1', and PT3), and whether differences were observed between ctDNA panels and tumor variants across these timepoints, would be a valuable addition to the supplemental data.

RESPONSE: Of the 94 patients with serial tumor tissues available for variant conservation analysis, 53 (56%) had tissues available at all three timepoints (PT0, PT1', and PT3). We have clarified this point in the revised Results and Supplementary Figure 9.

Similar results were observed in a subset analysis involving 53 (56%) patients with serial tumor samples for all three time points (Figure S9C).

Figure S9. Serial tumor mutation profiling of neoadjuvant therapy (NAT)-resistant tumors reveals the high conservation of patient-specific circulating tumor DNA (ctDNA) assay variants in the tissue over time. Mutation profiling by whole exome sequencing of matched tumor tissue samples was performed at pretreatment (PT0), on-treatment (week 3 or week 12, PT1'), and post-NAT at surgery (PT3) for a subset of patients with residual cancer burden (RCB)-II or RCB-III. **B.** Paired mutation profiling data were available for pretreatment (PT0) and on-treatment (PT1) tumors (n=76, left) and pretreatment (PT0) and post-NAT (PT3) tumors (n=71, right). The orange box plots show the distribution of the percentages of conserved somatic variants detected in paired tumor samples collected at PT0 and PT1, or at PT0 and PT3. The aqua box plots show the distribution of the percentages (left y-axis) or the number (right y-axis, up to 16 for each patient) of conserved personalized ctDNA assay variants detected in paired tumor samples collected at PT0 and PT1, and at PT0 and PT3. **C.** The same analysis (variant conservation) was restricted to patients with serial tumor samples for all three time points (n=53, 56%).

Minor points:

-Table S1 should include additional information such as: median follow up (and range), number of recurrences per receptor subgroup, and a breakdown of Grade 1 vs. Grade 2 (instead of grouping them together).

RESPONSE: We revised Table S1 to include the additional data requested.

Table S1. Patients and clinicopathologic characteristics. Patients were grouped by receptor subtypes: hormone receptor-positive/HER2-negative (HR+HER2-), triple-negative breast cancer (TNBC), and HER2-positive breast cancer. The survival endpoint of the study is distant recurrence-free survival (DRFS). For

categorical variables, the proportions were compared using Fisher's exact test. For continuous variables (age at screening), the distributions were compared using a t-test.

		HR+HER2- (n=300)	TNBC (n=237)	HER2+ (n=186)		
Follow-up information						
	Median (years)	4.6	4.72	4.76		
	Lower bound (years)	0.39	0.31	0.21		
	Upper bound (years)	11.51	11.05	8.58		
	No. of DRFS events	59	54	20	133	
Patient and tumor characteristics						
		n (%)	n (%)	n (%)	Total n (%)	p
Clinical T Stage						
						0.121
	T1	7 (2.4)	11 (4.7)	7 (3.8)	25 (3.5)	
	T2	183 (62.7)	158 (67.8)	130 (70.3)	471 (66.3)	
	T3	90 (30.8)	53 (22.7)	37 (20.0)	180 (25.4)	
	T4	12 (4.1)	11 (4.7)	11 (5.9)	34 (4.8)	
Clinical N Stage						
						<0.001
	Node-negative	96 (33.0)	123 (53.5)	82 (45.3)	301 (42.9)	
	Node-positive	195 (67.0)	107 (46.5)	99 (54.7)	401 (57.1)	
Grade						
						<0.001
	1	5 (2.1)	1 (0.6)	0 (0)	6 (1.1)	
	2	96 (41.0)	13 (7.8)	38 (30.6)	147 (28.1)	
	3	133 (56.8)	152 (91.6)	86 (69.4)	371 (70.8)	
MammaPrint						
						<0.001
	High 1	193 (64.3)	24 (10.1)	111 (59.7)	328 (45.4)	
	High 2	107 (35.7)	213 (89.9)	75 (40.3)	395 (54.6)	
Residual Cancer Burden (RCB)						
						<0.001
	RCB-0	55 (18.6)	87 (37.7)	99 (54.7)	241 (34.0)	
	RCB-I	30 (10.1)	35 (15.2)	15 (8.3)	80 (11.3)	
	RCB-II	130 (43.9)	75 (32.5)	50 (27.6)	255 (36.0)	
	RCB-III	81 (27.4)	34 (14.7)	17 (9.4)	132 (18.6)	
Age						
						0.758
	Mean (SD)	48.9 (11.0)	48.5 (11.9)	49.3 (9.7)	48.9 (11.0)	

-While a median follow-up of 4.7 years is sufficient in patients with HER2+ and TNBC, it is limited for those with HR+/HER2- disease where recurrence risk persists for decades. This should be acknowledged in the discussion.

RESPONSE: We have included this limitation in the revised **Discussion**.

A median follow-up of 4.7 years may be sufficient for the HER2-positive and TN subtypes; however, it may be insufficient for those with HR-positive/HER2-negative disease, where the risk of recurrence persists for decades. Hence, regular follow-up is ongoing in I-SPY2 to monitor for late recurrences in these patients.

-The authors should explain why mutation data was not available for all profiled patients with on treatment biopsies (Line 177-178).

RESPONSE: We have revised the **Methods** as follows:

Less than 50% of on-treatment biopsies contained enough tumor cells for WES. This might be because the biopsy missed the residual tumor or because the targeted clip used to guide the biopsy was in a tumor area responding to treatment.

-The variables in Figure S1B are not named consistently with Figure S1A

RESPONSE: The variable names are not consistent between Figure S1A and S1B. Thank you for pointing out this error.

-Discussion or inclusion of the number of DRFS events which have occurred in patients included in the Annals of Oncology and Cancer Cell manuscripts would be informative

RESPONSE: We have revised the **Methods** to include this information.

A total of 133 DRFS events were reported (**Table S1**). Of these, 12 occurred in patients from the original study²; 72 events were from patients in the second study, which was limited to the HER2-negative subtype¹; and 49 additional events involved patients in this study.

Reviewer #2 (Remarks to the Author):

We thank Reviewer #2 and their colleague for the insightful and helpful comments.

Reviewer #3 (Remarks to the Author)

In the manuscript titled “Circulating tumor DNA refines risk stratification of neoadjuvant therapy-resistant breast tumors,” Magbanua and colleagues investigate whether a tissue-informed circulating tumor DNA (ctDNA) assay can enhance risk stratification in patients with breast tumors resistant to neoadjuvant therapy (NAT) and predict treatment response. The study includes 723 patients with all breast cancer subtypes enrolled in the I-SPY2 trial. Among the evaluable patients, 300 (41%) had HR-positive/HER2-negative tumors, 237 (33%) had triple-negative (TN) tumors, and 186 (26%) had HER2-positive tumors. A tissue-informed ctDNA assay was applied at four time points during therapy. Key findings include: (1) ctDNA negativity was associated with significantly improved 3-year distant recurrence-free survival (DRFS) in patients with residual disease (RCB-II and III), and (2) ctDNA clearance by week 3 of neoadjuvant therapy predicted a favorable treatment response. In a subset of 94 residual tumors (45 HR-positive/HER2-negative, 38 TN, and 11 HER2-positive), the authors also assessed mutation conservation across serial tumor biopsies. They found a mutation conservation rate of <30% across tumor tissue samples compared to >90% in serial ctDNA samples. These findings suggest that low or undetectable ctDNA shedding and early ctDNA clearance may be characteristic of certain NAT-resistant tumors with inherently low metastatic potential. The manuscript is well written and addresses important, clinically relevant questions.

We thank Reviewer #3 for the insightful and helpful comments.

I believe it would be of interest to your journal’s readership. However, I recommend the following revisions:

Major

- In the section “ctDNA improves risk stratification of NAT-resistant tumors,” please include outcomes stratified by subtype, as differences in biology and ctDNA kinetics by subtype may yield important insights. If the numbers are too small to support meaningful conclusions, please clarify this in the text.

RESPONSE: We have revised the **Results** and added the following plots to **Figure S3** to address this question.

Similar results were observed across receptor subtypes (**Figure S3B**). Patients who remained persistently negative or cleared early at week 3 (T1) or 12 (T2) had better DRFS compared to those who cleared late post-NAT (T3) or did not clear ctDNA (all log-rank $p < 0.05$).

Figure S3. ctDNA dynamics refine risk stratification of breast tumors resistant to neoadjuvant therapy (NAT). Patients dynamics grouped patients. Kaplan-Meier estimates for 3-year DRFS rates in patients with NAT-resistant tumors defined as moderate (RCB-II) or extensive (RCB-III) residual cancer following NAT across receptor subtypes: (left) hormone receptor-positive/HER2-negative (HR+HER2-), (middle) triple-negative breast cancer (TNBC), and (right) HER2-positive (HER2+). The p-values for the survival curves were calculated using the log-rank test.

- Line 142: The sentence, “The proportion of patients who eventually achieved an RCB-0/RCB-I was the highest in this group in all receptor subtypes [HR-positive/HER2-negative: 42%, Chi-squared (χ^2) $p=0.02$]” — please verify whether this statement is accurate for the HR+/HER2- group. As written, 58% of these patients did not achieve RCB 0/I, which raises the question of whether this group indeed had the highest proportion.

RESPONSE: We revised the **Results** to clarify this point.

The proportion of patients who eventually achieved an RCB-0/RCB-I was highest among those who cleared ctDNA early at week 3 (T1) (**Figure 4B**). This pattern was consistently observed across all receptor subtypes [HR-positive/HER2-negative: 42%, Chi-squared (χ^2) $p=0.02$; TN: 82%, χ^2 $p<0.001$; HER2-positive: 82% χ^2 $p<0.001$].

- Consider adding if there were differences in ctDNA dynamics by tumor size, nodal status

RESPONSE: We have added **Supplementary Figures S6** and revised the **Results**.

Significant associations were observed between early clearance at week 3 (T1) and a favorable response to NAT, regardless of clinical T stage or nodal status (all χ^2 $p<0.05$; **Figure S6**)

Figure S6. The association between early circulating tumor DNA (ctDNA) dynamics and residual cancer burden (RCB) class by clinical T and N stages. Patients were grouped by early ctDNA dynamics: persistent negative, cleared at T1 (week 3) or T2 (week 12), and no clearance at T2 and stratified by **A.** clinical T stage (T1/T2 and T3/T4), and **B.** clinical N stage (node-positive and node-negative). Bar plots showing the proportion of RCB classes in each early ctDNA dynamics group. The percentages may not add up to 100% due to rounding. The p-values were calculated using the Chi-squared test.

- Limitations: describe limitations of Signatera, including the sensitivity, describe if any tumor informed assays could not be performed due to the lack of tissue, and how this could impact clinical implementation (as patients are more likely to lack available tissue if they are not enrolled in a clinical trial).

RESPONSE: We have revised the **Discussion** to describe these limitations.

Tumor-informed ctDNA tests require tumor tissue for sequencing to identify personalized ctDNA assay variants⁶. Tumor samples may be unavailable, inadequate, or of poor quality, which presents a recognized limitation for this type of test. This could impact clinical implementation, particularly in settings outside of clinical

trials, where tumor tissue may be less readily available. However, since the ctDNA assay can be designed based on various sources of tissues, including diagnostic biopsy or surgically resected primary or metastatic tissue^{7,8}, as well as on-treatment and post-NAT tumors, as suggested by the findings of this study, tumor-informed ctDNA tests are feasible for the majority of patients. Another known limitation of tumor-informed tests, especially if designed using pretreatment tumors, is that they will miss mutations that emerge during or after therapy.

- Supplementary table 2: please add more detail about the treatments received

RESPONSE: We have added more details about treatments received to the supplementary table.

Table S3. Types of treatments in I-SPY2. Patients were assigned to different arms in the I-SPY2 trial. The control arm and the treatment arms, for which ctDNA analysis has been completed for all evaluable patients, were included in the analysis. These arms were grouped into four treatment types (see **Methods**).

Treatment Type	N by Agent	N by Subtype	Total
Control	Paclitaxel (N=161)	HR+HER2- (n=88) TNBC (n=73)	161
Small molecule inhibitor-containing	Irinotecan + Talazoparib (n=49) ABT 888 + Carboplatin (n=58) MK-2206 (n=39)	HR+HER2- (n=66) TNBC (n=80)	146
ICI-containing	Anti-PD-1 4 cycles (n=46) Anti-PD-1 8 cycles (n=56) Durvalumab + Olaparib (n=52)	HR+HER2- (n=105) TNBC (n=49)	154
HER2-targeted	Pertuzumab + Trastuzumab (n=111) Trastuzumab (n=8) T-DM1 + Pertuzumab (n=42)	HER2+ (n=161)	161

Minor

- In the introduction, please clarify that pathologic complete response (pCR) is a strong predictor of long-term outcomes in HER2-positive and triple-negative breast cancer, but that its prognostic value is less robust in HR-positive/HER2-negative disease.

RESPONSE: We have revised the **Introduction** as follows:

A pathologic complete response (pCR) to NAT, marked by the absence of residual cancer burden (RCB-0) in the breast and regional lymph nodes, is a strong predictor of favorable long-term outcomes, especially in HER2-positive and triple-negative (TN) breast cancer subtypes; however, its prognostic value is less robust in hormone receptor (HR)-positive/HER2-negative disease⁹.

- Consider adding background in the introduction about the median conservation rate of mutations across paired tumor samples and its potential clinical implications.

RESPONSE: We have revised the **Introduction** as follows:

There are various types of ctDNA assays designed for different applications^{10,11}. The main categories include tumor-agnostic and tumor-informed tests⁶. Tumor-agnostic methods do not require tumor sequencing and utilize the same fixed panel of assays in every case to detect common cancer mutations in the blood¹². In contrast, tumor-informed methods need prior knowledge of existing tumor mutations; thus, sequencing of the tumor tissue is a prerequisite to “inform” the subsequent design of patient-specific ctDNA assays¹². The sequence information, however, only provides a snapshot of the tumor’s molecular profile at a particular point in time. Changes in the mutational landscape during tumor evolution [5]—whether due to clonal selection, therapeutic pressure, or spatial heterogeneity—can result in the emergence of mutations that are not present in the profiled tumor sample.

Here, we analyzed serial tumor mutational profiling data from matched NAT-resistant tumors to investigate whether changes in the mutational landscape during NAT impacted the detection of patient-specific ctDNA assay variants.

- Line 158: Please spell out "ICI" (immune checkpoint inhibitor) at first mention for clarity.

RESPONSE: We have spelled ICI for clarity.

To examine the predictive value of ctDNA across treatment types, we categorized treatments received by patients with HR-positive/HER2-negative and TN disease into three categories: 1. Paclitaxel (P) alone, 2. P + small molecule inhibitor (SMI), and 3. P + immune checkpoint inhibitor (ICI).

- Figure 2B: consider adding graphs by subtype

RESPONSE: We have revised **Figure 2** to add a plot by subtype.

There was no significant difference in the pretreatment (T0) ctDNA positivity rates observed between the HR-positive/HER2-negative and the HER2-positive groups (76% versus 77%, proportion test $p=0.75$). However, 3 weeks after initiation of treatment (T1), the ctDNA positivity rate in the HR-positive/HER2-negative subtype was significantly higher compared to that of the HER2-positive group (45% versus 25%, proportion test $p<0.001$). The ctDNA positivity rate remained the lowest in the HER2-positive group at 12 weeks (T2) and post-NAT before surgery (T3) (**Figure 2B**).

Figure 2B. Line plot showing the proportion of ctDNA-positive patients from T0 to T3, stratified by receptor subtype.

- Figures 3B and 3C: ctDNA- appears to have a strongest correlations with outcomes vs RCBII vs III, this is very interesting and could be included in the text

RESPONSE: We have revised the **Results** as follows:

These results demonstrate that ctDNA negativity at pretreatment (T0) and post-NAT before surgery (T3) significantly correlated with reduced risk of metastatic recurrence, even in patients with RCB-II or RCB-III after NAT.

- If available, it may be interesting to add data on HER2+/HR+ vs HER2+/HR-, there is minimal literature in this setting and this could strengthen the paper

RESPONSE: We agree that comparing the clinical significance of ctDNA in HR+HER2+ vs. HR-HER2+ would strengthen the paper. However, our current dataset has a limited sample size, which does not allow for meaningful comparisons between these two subtypes. Efforts are ongoing to expand our dataset to include an additional ~700 I-SPY2 patients.

We have revised the **Discussion** as follows:

Current efforts in I-SPY2 involve expanding the study to an additional ~700 patients, which could facilitate comparisons of the clinical significance of ctDNA among groups within the HER2-positive (HR-negative/HER2-positive vs. HR-positive/HER2-positive) and histological (invasive lobular carcinoma vs. invasive ductal carcinoma) subtypes.

Reviewer #4 (Remarks to the Author)

The authors present data from a large and well-designed longitudinal study of circulating tumor DNA (ctDNA) in early stage breast cancer patients undergoing neoadjuvant therapy. The study includes a substantial cohort (>700 patients) with serial sampling at four timepoints, and explores associations between ctDNA dynamics, pathological response, and recurrence-free survival. The methodology is rigorous and clearly described. The results convincingly demonstrate the potential of serial ctDNA profiling as a clinically informative tool in this setting. The only major limitation is the final analysis concerning ctDNA assay variant conservation over time, which would benefit from further clarification and refinement (see Major Point 1 below).

We thank Reviewer #4 for the insightful and helpful comments.

Major Points

1. Analysis of ctDNA Assay Variant Conservation (Figure 6)

The manuscript reports low median conservation rates of 24.8% (range: 4.8–44.9%) between paired PT0 and PT1' tumors (Figure S5A), and 23.2% (range: 2.8–40.5%) between PT0 and PT3 tumors (Figure S5B), suggesting changes in the tumor mutational landscape.

However, the low concordance between tumor samples at different timepoints may reflect sampling variability and tumor heterogeneity—particularly the presence of spatially distributed subclones—rather than true biological shifts in the mutational landscape.

RESPONSE: While we cannot completely rule out spatial variability and tumor heterogeneity as factors contributing to the observed changes in the mutational landscape, measures were implemented to reduce sampling variability. We have updated the **Methods** section to clarify this.

To minimize the impact of low tumor content on variant allele fraction (VAF) interpretation (subclonal vs. clonal inference)¹³, only biopsy cores with high-tumor cellularity ($\geq 30\%$) were used for WES. Additionally, quality control measures, such as pathology review, were performed to identify the most representative regions of the tumor for sequencing.

The authors should clarify the specific question this analysis aims to address. If the goal is to evaluate variant conservation between tumor and ctDNA assays, additional context is needed.

RESPONSE: We have revised the **Introduction** to provide context for the variant conservation analysis.

There are various types of ctDNA assays designed for different applications^{10,11}. The main categories include tumor-agnostic and tumor-informed tests⁶. Tumor-agnostic methods do not require tumor sequencing and utilize the same fixed panel of assays in every case to detect common cancer mutations in the blood¹². In contrast, tumor-informed methods need prior knowledge of existing tumor mutations; thus, sequencing of the tumor tissue is a prerequisite to “inform” the subsequent design of patient-specific ctDNA assays¹². The sequence information, however, only provides a snapshot of the tumor’s molecular profile at a particular point in time. Changes in the mutational landscape during tumor evolution [5]—whether due to clonal selection, therapeutic pressure, or spatial heterogeneity—can result in the emergence of mutations that are not present in the profiled tumor sample.

Here, we analyzed serial tumor mutational profiling data from matched NAT-resistant tumors to investigate whether changes in the mutational landscape during NAT impacted the detection of patient-specific ctDNA assay variants.

Specifically, the authors should:

* Define what they mean by "variant conservation" and how it is calculated.

RESPONSE: We have revised the **Methods** to clarify the definition of variant conservation.

A variant was deemed “conserved” if it was detected in matched serial tumor samples, i.e., shared variants. The percentage of variant conservation was calculated as the number of shared variants between paired tissue samples [pretreatment (PT0) versus on-treatment (PT1’) tumors or pretreatment (PT0) versus post-NAT (PT3) tumors] divided by the number of variants detected in pretreatment (PT0) tumors.

* Compare the total number of variants detected in tumor vs. ctDNA assays.

RESPONSE: We selected only 16 patient-specific variants to track ctDNA in plasma samples, regardless of the number of variants detected in the pretreatment tumor. We have also revised the **Results** as follows:

WES analysis of pretreatment tumors (PT0) in 94 patients with RCB-II or RCB-III detected a median of 603 variants (range 264-1509). There was no significant difference (Kruskal-Wallis $p=0.4392$) when compared to the median number of variants detected in 76 on-treatment (PT1’) tumors (median 588, range 265-6360) and 71 post-NAT (PT3) tumors (median 595, range 277-7468). We then checked for the presence of 16 ctDNA assay variants selected from WES of PT0 in the matched serial tumors. A median of 16 and 15 variants were detected in PT1’ and PT3 tumors, respectively.

* Examine the distribution of variant allele frequencies (VAFs) in both tumor and plasma samples.

RESPONSE: We performed this analysis and have revised the manuscript as follows:

Results

We compared the distribution of variant allele frequencies (VAF) of the patient-specific ctDNA variants in the tumor tissue with those in the plasma (ctDNA) at pretreatment (T0, **Figure S9A**). We observed significantly higher VAFs in tissue compared to plasma (Wilcoxon $p<0.001$).

Figure S9A. Box plots comparing the distribution of the variant allele frequencies (VAF) of the patient-specific ctDNA assays in the tumor tissue versus those in the plasma (ctDNA) at pretreatment (T0). The p-value was calculated using the Wilcoxon signed-rank test.

Discussion

Comparison of the distribution of VAFs of patient-specific ctDNA assays in tumor tissue versus plasma (ctDNA) yielded results consistent with findings from several studies¹⁴⁻¹⁶. Differences in VAFs between tumor tissue and plasma (ctDNA) can be due to several technical (e.g., preanalytical conditions and the ctDNA assay used) and biological factors. VAF in the tissue reflects the proportion of altered loci in the tumor, admixed with normal DNA from stromal cells (when present) and tumor subclones that do not carry the alteration. In contrast, VAF in plasma reflects the proportion of tumor-derived molecules (ctDNA) in the background of cell-free DNA¹⁷. Key contributors to these differences include variable ctDNA shedding, high background of normal cell-free DNA (cfDNA), and tumor status. The rate of ctDNA in plasma is influenced by tumor size, location, vascularization, and cell turnover and death (e.g., apoptosis or necrosis)¹⁸. In addition, the plasma contains an admixture of

cfDNA from both tumor and normal cells (predominantly of hematopoietic origin), which can dilute the concentration of the ctDNA¹⁹. The concentration of ctDNA in the plasma can also reflect the current biological state of the tumor, which may change depending on treatment response^{1,2} and disease status⁶.

* Assess whether restricting the tumor analysis to high-VAF (likely clonal) mutations alters the degree of concordance.

RESPONSE: We have revised the **Results** and **Supplementary Figure 9** as follows:

We found that restricting the tumor analysis to mutations with high VAF (>median VAF) increased the degree of concordance (**Figure S9D**). The percentage of variant conserved between PT0 versus PT1' was 24.9% for all variants compared to 29.9% when including only high-VAF variants (Wilcoxon $p < 0.001$). Similar results were observed for PT0 versus PT3 (all variants: 23.3% versus high-VAF variants: 28.9%, Wilcoxon $p < 0.001$). All ctDNA assay variants exhibit high VAFs (>median VAF in the tumor), which is a criterion for inclusion in the panel of 16 patient-specific assays; thus, the restriction does not impact the level of concordance.

Figure S9. Serial tumor mutation profiling of neoadjuvant therapy (NAT)-resistant tumors reveals the high conservation of patient-specific circulating tumor DNA (ctDNA) assay variants in the tissue over time. Mutation profiling by whole exome sequencing of matched tumor tissue samples was performed at pretreatment (PT0), on-treatment (week 3 or week 12, PT1'), and post-NAT at surgery (PT3) for a subset of patients with residual cancer burden (RCB)-II or RCB-III. Paired mutation profiling data were available for pretreatment (PT0) and on-treatment (PT1) tumors ($n=76$, **left**) and pretreatment (PT0) and post-NAT (PT3) tumors ($n=71$, **right**). The orange box plots show the distribution of the percentages of conserved somatic variants detected in paired tumor samples collected at PT0 and PT1, or at PT0 and PT3. The aqua box plots show the distribution of the percentages (left y-axis) or the number (right y-axis, up to 16 for each patient) of conserved personalized ctDNA assay variants detected in paired tumor samples collected at PT0 and PT1, and at PT0 and PT3. The same analysis (variant conservation) was restricted to mutations with high VAF (>median VAF) (green boxes). The p -value was calculated using the Wilcoxon rank sum test.

These additions would strengthen the interpretation and provide greater insight into the biological and technical factors influencing variant detection over time.

2. Multivariate Statistical Analysis

The study would benefit from a multivariate analysis incorporating ctDNA status, tumor stage, RCB, and other available clinical prognostic factors. This would allow a more robust evaluation of the independent prognostic value of ctDNA and facilitate comparison with established clinical predictors.

RESPONSE: We have added univariable and multivariable analyses to evaluate the prognostic value of ctDNA in conjunction with other established clinicopathologic predictors of metastatic recurrence.

We have revised the manuscript as follows:

Methods

To evaluate the prognostic performance of ctDNA, RCB, and other clinicopathologic variables for predicting DRFS, we fitted Cox proportional hazards models using the `coxph` function from the survival R package. Univariable (**Table S2**) and multivariable Cox regression analyses were performed to adjust for potential confounders.

Results

ctDNA independently predicts metastatic recurrence

Next, we assessed the prognostic significance of ctDNA, along with other clinicopathological factors, including clinical T and N stages, grade, MammaPrint score, and RCB. DRFS data were available for 712 patients, of whom 133 (19%) experienced a DRFS event. The median follow-up was 4.7 years.

Multivariable Cox regression analysis, which included clinicopathologic variables that were statistically significant in univariable analyses (**Table S2**), showed that ctDNA positivity at pretreatment (T0) is an independent predictor of metastatic recurrence [adjusted hazard ratio (adj HzR)=4.40, 95% confidence interval (CI) 1.91–10.16, Wald $p=0.001$] (**Figure S1A**). The same analysis, performed in parallel, also identified ctDNA positivity after NAT before surgery (T3) as an independent prognostic factor for poor outcomes [adj HzR=5.20, 95% CI 3.24–8.35, Wald $p<0.001$] (**Figure S1B**).

We then assessed the prognostic significance of ctDNA dynamics (timing of ctDNA clearance). Patients were divided into 5 groups: those who remained persistently ctDNA-negative, those who cleared ctDNA at T1 (week 3), T2 (week 12), or T3 (post-NAT before surgery), and those who did not clear ctDNA (no clearance at T3). Late and no clearance were both significant independent negative prognostic factors for DRFS [adj HzR=6.88, 95% CI 2.37–19.98, Wald $p<0.001$; adj HzR=16.50, 95% CI 5.67–47.98, Wald $p<0.001$, respectively] (**Figure S1C**).

Table S2. Univariate Cox regression analysis. Correlation of clinicopathological variables with distant recurrence-free survival (DRFS). Abbreviation: CI – confidence interval; HR – hormone receptor; RCB – residual cancer burden; TNBC – triple-negative breast cancer

Variable*	Factor	Reference	Hazard ratio	Lower 95% CI	Upper 95% CI	Wald p
ctDNA	ctDNA+ at T0	ctDNA-	5.50	2.43	12.49	<0.0001
	ctDNA+ at T3		10.36	6.92	15.50	<0.0001
Clinical T stage	T3/T4	T1/T2	2.17	1.53	3.07	<0.0001
Clinical N stage	Node+	Node-	1.47	1.02	2.12	0.0373
Grade	3	1/2	1.31	0.82	2.07	0.2557
MammaPrint	High 2	High 1	1.25	0.88	1.76	0.2066
Receptor Subtype	HR+HER2-	HER2+	0.53	0.32	0.88	0.0143
	TNBC		1.22	0.84	1.76	0.2944
RCB class	RCB-I	RCB-0	1.96	0.89	4.33	0.0939
	RCB-I		3.20	1.82	5.62	0.0001
	RCB-III		7.63	4.35	13.37	<0.0001
Variable*	Factor	Reference	Hazard ratio	Lower 95% CI	Upper 95% CI	Wald p
ctDNA dynamics	Cleared at T1	Persistent ctDNA-	1.92	0.64	5.75	0.2428
	Cleared at T2		2.47	0.78	7.88	0.1260
	Cleared at T3		8.96	3.11	25.85	<0.0001
	No clearance at T3		30.00	10.60	84.87	<0.0001
Clinical T stage	T3/T4	T1/T2	2.03	1.33	3.11	0.0011
Clinical N stage	Node+	Node-	1.87	1.17	2.96	0.0082
Grade	3	1/2	1.11	0.65	1.90	0.6906
MammaPrint	high 2	high 1	1.27	0.83	1.94	0.2702
Receptor Subtype	HR+HER2-	HER2+	0.61	0.33	1.12	0.1090
	TNBC		1.28	0.81	2.03	0.2872
RCB class	RCB-I	RCB-0	2.29	0.92	5.70	0.0741
	RCB-I		3.30	1.66	6.56	0.0007
	RCB-III		9.15	4.64	18.02	<0.0001

*Variables in bold are statistically significant in univariate Cox regression analysis.

Figure S1. Circulating tumor DNA (ctDNA) is an independent predictor of distant recurrence-free survival (DRFS) in patients with high-risk early-stage breast cancer who received neoadjuvant therapy (NAT). Multivariable Cox regression analyses included clinicopathologic variables that were statistically significant in univariable analyses (Table S2) to evaluate the prognostic value of **A.** ctDNA positivity at pretreatment (T0), **B.** post-NAT before surgery (T3), as well as **C.** ctDNA dynamics (timing of ctDNA clearance). Patients were grouped by ctDNA dynamics, including persistent ctDNA-negative, ctDNA cleared at T1 (week 3), T2 (week 12), or T3 (post-NAT before surgery), or no ctDNA clearance post-NAT before surgery. Cox regression analysis was used to estimate hazard ratios (HzR) and 95% confidence intervals (CI). The p-values were calculated using the Wald test

Minor Point

The authors are encouraged to explore and report any differences or similarities in ctDNA dynamics and predictive value between ductal and lobular breast cancer subtypes within their cohort, if data allow. This could provide useful insight into subtype-specific applicability of ctDNA monitoring.

RESPONSE: We agree that comparing ctDNA between invasive lobular carcinoma and invasive ductal carcinoma could provide helpful insights into the subtype-specific applicability of ctDNA monitoring. We have a manuscript in preparation to address this critical question separately.

We have revised the **Discussion** as follows:

Current efforts in I-SPY2 involve expanding the study to an additional ~700 patients, which could facilitate comparisons of the clinical significance of ctDNA among groups within the HER2-positive (HR-negative/HER2-positive vs. HR-positive/HER2-positive) and histological (invasive lobular carcinoma vs. invasive ductal carcinoma) subtypes.

REFERENCES

- 1 Magbanua, M. J. M. *et al.* Clinical significance and biology of circulating tumor DNA in high-risk early-stage HER2-negative breast cancer receiving neoadjuvant chemotherapy. *Cancer Cell*, doi:<https://doi.org/10.1016/j.ccell.2023.04.008> (2023).
- 2 Magbanua, M. J. M. *et al.* Circulating tumor DNA in neoadjuvant-treated breast cancer reflects response and survival. *Ann Oncol* **32**, 229-239, doi:10.1016/j.annonc.2020.11.007 (2021).
- 3 Harrell, F. E., Jr., Lee, K. L., Califf, R. M., Pryor, D. B. & Rosati, R. A. Regression modelling strategies for improved prognostic prediction. *Stat Med* **3**, 143-152, doi:10.1002/sim.4780030207 (1984).
- 4 Schroder, M. S., Culhane, A. C., Quackenbush, J. & Haibe-Kains, B. survcomp: an R/Bioconductor package for performance assessment and comparison of survival models. *Bioinformatics* **27**, 3206-3208, doi:10.1093/bioinformatics/btr511 (2011).
- 5 Symmans, W. F. *et al.* Long-Term Prognostic Risk After Neoadjuvant Chemotherapy Associated With Residual Cancer Burden and Breast Cancer Subtype. *J Clin Oncol* **35**, 1049-1060, doi:10.1200/JCO.2015.63.1010 (2017).
- 6 Medford, A. J. *et al.* Molecular Residual Disease in Breast Cancer: Detection and Therapeutic Interception. *Clin Cancer Res* **29**, 4540-4548, doi:10.1158/1078-0432.CCR-23-0757 (2023).
- 7 Abdelrahim, M. *et al.* Feasibility of Personalized and Tumor-Informed Circulating Tumor DNA Assay for Early Recurrence Detection in Patients With Hepatocellular Carcinoma. *JCO Precis Oncol* **9**, e2400934, doi:10.1200/PO-24-00934 (2025).
- 8 Foldi, J. *et al.* Personalized Circulating Tumor DNA Testing for Detection of Progression and Treatment Response Monitoring in Patients With Metastatic Invasive Lobular Carcinoma of the Breast. *JCO Precis Oncol* **9**, e2400577, doi:10.1200/PO-24-00577 (2025).
- 9 Spring, L. M. *et al.* Pathologic Complete Response after Neoadjuvant Chemotherapy and Impact on Breast Cancer Recurrence and Survival: A Comprehensive Meta-analysis. *Clin Cancer Res* **26**, 2838-2848, doi:10.1158/1078-0432.CCR-19-3492 (2020).
- 10 Amato, O., Giannopoulou, N. & Ignatiadis, M. Circulating tumor DNA validity and potential uses in metastatic breast cancer. *NPJ Breast Cancer* **10**, 21, doi:10.1038/s41523-024-00626-6 (2024).
- 11 Nader-Marta, G. *et al.* Circulating tumor DNA for predicting recurrence in patients with operable breast cancer: a systematic review and meta-analysis. *ESMO Open* **9**, 102390, doi:10.1016/j.esmoop.2024.102390 (2024).
- 12 Panet, F. *et al.* Use of ctDNA in early breast cancer: analytical validity and clinical potential. *NPJ Breast Cancer* **10**, 50, doi:10.1038/s41523-024-00653-3 (2024).
- 13 Sisoudiya, S. D. *et al.* Tissue-based genomic profiling of 300,000 tumors highlights the detection of variants with low allele fraction. *NPJ Precis Oncol* **9**, 190, doi:10.1038/s41698-025-00991-w (2025).
- 14 Vasseur, D. *et al.* Genomic landscape of liquid biopsy mutations in TP53 and DNA damage genes in cancer patients. *NPJ Precis Oncol* **8**, 51, doi:10.1038/s41698-024-00544-7 (2024).
- 15 Leenanitikul, J. *et al.* Concordance between whole exome sequencing of circulating tumor DNA and tumor tissue. *PLoS One* **18**, e0292879, doi:10.1371/journal.pone.0292879 (2023).
- 16 Kim, J. J. *et al.* Circulating Tumor DNA Reflects Histologic and Clinical Characteristics of Various Lymphoma Subtypes. *Cancer Res Treat* **56**, 314-323, doi:10.4143/crt.2023.667 (2024).

- 17 Kalashnikova, E. *et al.* Correlation between variant allele frequency and mean tumor molecules with tumor burden in patients with solid tumors. *Mol Oncol* **18**, 2649-2657, doi:10.1002/1878-0261.13557 (2024).
- 18 Stejskal, P. *et al.* Circulating tumor nucleic acids: biology, release mechanisms, and clinical relevance. *Mol Cancer* **22**, 15, doi:10.1186/s12943-022-01710-w (2023).
- 19 Mattox, A. K. *et al.* The Origin of Highly Elevated Cell-Free DNA in Healthy Individuals and Patients with Pancreatic, Colorectal, Lung, or Ovarian Cancer. *Cancer Discov* **13**, 2166-2179, doi:10.1158/2159-8290.CD-21-1252 (2023).

Below, in blue font, are our detailed responses to each point.

REVIEWERS' COMMENTS

Reviewer #1 (Remarks to the Author):

The authors have provided a comprehensive response to the initial reviews. The manuscript has been substantially improved. A few minor points remain:

RESPONSE: The authors thank Reviewer #1 for the kind words and helpful suggestions.

Consider incorporating Figure S5 into the main manuscript (e.g., as part of Figure 3). Doing so would not only highlight the prognostic value of the findings but also clearly present the associated performance metrics, which would enhance the impact of the results.

RESPONSE: We have added Figure S5 to the main manuscript.

Language that implies judgement on the patient, ie patient “achieved” (or conversely didn’t achieve), or “patient cleared” should be avoided. Instead, neutral language that captures the treatment effect should be used. Eg. pCR was observed, ctDNA clearance occurred or was observed, or the treatment resulted in ctDNA clearance etc

RESPONSE: We apologize for this oversight. We (including Amy L. Delson, co-author and patient advocate) have revised the manuscript to incorporate non-judgmental language in the manuscript.

The authors note that additional patients are currently being evaluated as part of an expanded analysis. Given reported whole-genome based Signatera Assays, the authors should be encouraged to compare the clinical performance of these methods.

RESPONSE: We have revised the Discussion as follows:

Studies involving the retesting of plasma samples to determine whether expanding the panel of patient-specific ctDNA variant assay (e.g., including mutations that emerge in on-treatment and post-NAT tumors or using the Signatera™ Genome test with 64 targets) could improve ctDNA detection rates are being planned.

Reviewer #2 (Remarks to the Author):

RESPONSE: We thank Reviewer #2 for their time reviewing the manuscript with Reviewer #1.

Reviewer #3 (Remarks to the Author):

In “Circulating tumor DNA refines risk stratification of neoadjuvant therapy-resistant breast tumors,” Magbanua and colleagues evaluate a tissue-informed circulating tumor DNA (ctDNA) assay in 723 I-SPY2 participants with breast tumors resistant to neoadjuvant therapy (NAT). Patients included HR-positive/HER2-negative (41%), triple-negative (33%), and HER2-positive (26%) subtypes. ctDNA was measured at four time points. ctDNA negativity was linked to improved 3-year distant recurrence-free survival in patients with residual disease, and clearance by week 3 predicted favorable response. In 94 residual tumors, mutation conservation was <30% in tissue but >90% in serial ctDNA, suggesting ctDNA better reflects tumor evolution. These findings indicate that low/undetectable ctDNA and early clearance may characterize NAT-resistant tumors with low metastatic potential. The manuscript is well written and clinically relevant.

The authors did an excellent job addressing my prior comments. The paper is now clearer, and with the additional information about subtypes, readers can better understand ctDNA dynamics.

RESPONSE: We thank Reviewer #3 for the kind words and helpful editorial comment.

My only minor editorial comment is regarding Figure S3B: the time points shown above the subtypes may be confusing since they overlap with the x-axis. Shifting them slightly could improve clarity.

RESPONSE: We have revised Figure S3B to improve clarity.

Reviewer #4 (Remarks to the Author):

The authors have addressed all my concerns, thanks!

One minor comment, Table S2 appears to have a typo ("RCB I" repeated twice).

RESPONSE: We thank Reviewer #4 for their time reviewing the manuscript. We have corrected the error.